# Gene body DNA hydroxymethylation restricts the magnitude of transcriptional changes during aging

James R. Occean [1], Na Yang [1], Yan Sun[2], Marshall S. Dawkins[1], Rachel Munk[1], Cedric Belair [1], Showkat Dar [1], Carlos Anerillas [1], Lin Wang[1], Changyou Shi[1], Christopher Dunn [3], Michel Bernier [4], Nathan L. Price[4], Julie S. Kim[2], Chang-Yi Cui[1], Jinshui Fan [5], Moitrayee Bhattacharyya [6], Supriyo De [5], Manolis Maragkakis [1], Rafael de Cabo [4], Simone Sidoli [2] & Payel Sen [1] ✉

DNA hydroxymethylation (5hmC), the most abundant oxidative derivative of DNA methylation, is typically enriched at enhancers and gene bodies of transcriptionally active and tissue-specific genes. Although aberrant genomic 5hmC has been implicated in age-related diseases, its functional role in aging remains unknown. Here, using mouse liver and cerebellum as model organs, we show that 5hmC accumulates in gene bodies associated with tissue-specific function and restricts the magnitude of gene expression changes with age. Mechanistically, 5hmC decreases the binding of splicing associated factors and correlates with age-related alternative splicing events. We found that various age-related contexts, such as prolonged quiescence and senescence, drive the accumulation of 5hmC with age. We provide evidence that this age-related transcriptionally restrictive function is conserved in mouse and human tissues. Our findings reveal that 5hmC regulates tissue-specific function and may play a role in longevity.

Aging is characterized by progressive deterioration of physiological function and is the main risk factor for morbidity and mortality. Aging tissues display altered transcriptional landscapes[1–4], consistent with an important role for gene regulation in aging. Indeed, epigenetic alterations, which involve mechanisms that shape chromatin architecture and modulate gene expression patterns, are a cardinal feature of aging[5–7] and age-related diseases, including cancer[8], cardiovascular disease[9], neurodegeneration[10], obesity[11], and type 2 diabetes[11]. Due to the reversible nature of epigenetic marks, they offer promising therapeutic targets for ameliorating age-related functional decline.

In this study, we investigated the role of DNA hydroxymethylation (5-hydroxymethylcytosine, 5hmC), an understudied epigenetic mark, in aging. 5hmC is the most abundant oxidative derivative of DNA

methylation (5-methylcytosine, 5mC), a better-known biomarker of age[12–14]. The oxidation of 5mC to 5hmC is mediated by the ten-eleven translocation (TET) Fe$^{2+}$ 2-oxoglutarate dioxygenase family of enzymes[15,16]. Several factors may contribute to the formation of 5hmC. For example, (i) reactive oxygen species (ROS) can form 5hmC via abstraction of an H-atom from the methyl group of 5mC[17]; (ii) vitamin C can induce DNA demethylation and increase 5hmC by acting as a cofactor for TET enzymes, facilitating the reduction of Fe$^{3+}$ to Fe$^{2+}$ [18,19]; (iii) prolonged quiescence can also increase the level of genomic 5hmC over time. The latter is hinted by findings showing that DNA replication (such as in cancer) dilutes 5hmC[20–22], and is further supported by the present study. Additionally, TET proteins are more prone to oxidizing 5mC to 5hmC rather than further oxidizing 5hmC to its oxidative

[1]Laboratory of Genetics and Genomics, National Institute on Aging, NIH, Baltimore, MD, USA. [2]Department of Biochemistry, Albert Einstein School of Medicine, Bronx, NY, USA. [3]Flow Cytometry Unit, National Institute on Aging, NIH, Baltimore, MD, USA. [4]Translational Gerontology Branch, National Institute on Aging, NIH, Baltimore, MD, USA. [5]Computational Biology and Genomics Core, Laboratory of Genetics and Genomics, National Institute on Aging, NIH, Baltimore, MD, USA. [6]Department of Pharmacology, Yale University, New Haven, CT, USA. ✉e-mail: payel.sen@nih.gov

product, 5-formylcytosine (5fC), leading to the accumulation of 5hmC over time[23]. Moreover, activity of the TET enzymes is dependent on α-ketoglutarate, a metabolite generated in the tricarboxylic acid cycle and previously reported to promote longevity in various organisms[24–26]. Recent work has also shown that TET enzymes, namely TET1 and TET2, are required for the age-related protective and regenerative effects of OSK (*Oct4, Sox2, and Klf4*)-induced partial reprogramming in mouse retinal ganglion cells[27]. These studies strongly suggest a potential function for 5hmC in aging.

5hmC was formerly regarded as an intermediary step in the DNA demethylation pathway. Importantly, previously established bisulfite-based techniques (e.g., methylation arrays, reduced representation bisulfite sequencing, whole-genome bisulfite sequencing, etc.) cannot distinguish 5mC from 5hmC[28], thereby confounding interpretations and potentially minimizing the contribution of this modification. Recent studies have revealed that 5hmC interacts with specific proteins (or readers) unique from those associated with 5mC[29]. Certain 5hmC interactors, such as UHRF2 (Ubiquitin Like With PHD And Ring Finger Domains 2), exhibited tissue-specificity and were dynamically regulated during cellular differentiation[29]. Concomitantly, multi-tissue studies have documented the presence of 5hmC at enhancer regions and gene bodies of tissue-specific and transcriptionally active genes[30–36], suggesting an active regulatory role for the modification. Additionally, findings from another study indicates a role for 5hmC in averting lung inflammation by preventing inappropriate intragenic transcription in smooth muscle cells[37].

Here, we surveyed global levels of 5hmC in multiple young and old organs. We report that 5hmC accumulates in the aged liver, partly due to prolonged quiescence. Using mouse liver and cerebellum as model organs, we show that gene body 5hmC plays a critical role in restricting the magnitude of transcriptional changes during aging, especially over tissue-specific genes. We validate this finding in published 5hmC datasets obtained from post-mortem human samples. Lastly, we show that this transcriptionally restrictive function might be detrimental in the response to stress, and that reduction of 5hmC under stress conditions is important for longevity-promoting interventions.

## Results

### Global accumulation of 5hmC in the aging liver

We initiated this study by performing a global assessment of 5mC and 5hmC in genomic DNA (gDNA) of young (~2–5 months) and old (~18–20 months, $n = 4$ biological replicates per age group) C57BL/6JN mice tissues of both sexes, including brain (cortex, cerebellum, hippocampus), heart, lung, liver, spleen, kidney, and muscle (Fig. 1A, Supplementary Dataset 1 & 2). As reported previously, brain showed the highest absolute levels of 5hmC (~0.51%)[38] by nano liquid chromatography tandem mass spectrometry (nLC-MS/MS) followed by muscle (0.28%), liver (0.24%), kidney (0.17%), heart (0.17%), lung (0.15%), and spleen (0.11%). Of note, the brain and heart are largely post-mitotic, and the liver, kidney and lung are mildly proliferative upon injury, while the spleen is a site of active proliferation; thus, levels of 5hmC, to some degree, correlate with tissue proliferative capacity. There were no significant differences in 5mC levels between old and young for any of the tissues measured; however, a significant increase in 5hmC levels was detected between the old and young liver (mean difference = 0.24%). The global increase of 5hmC with age without detectable differences in 5mC is concordant with previously reported LC-MS/MS results from mouse liver tissue[39] and does not

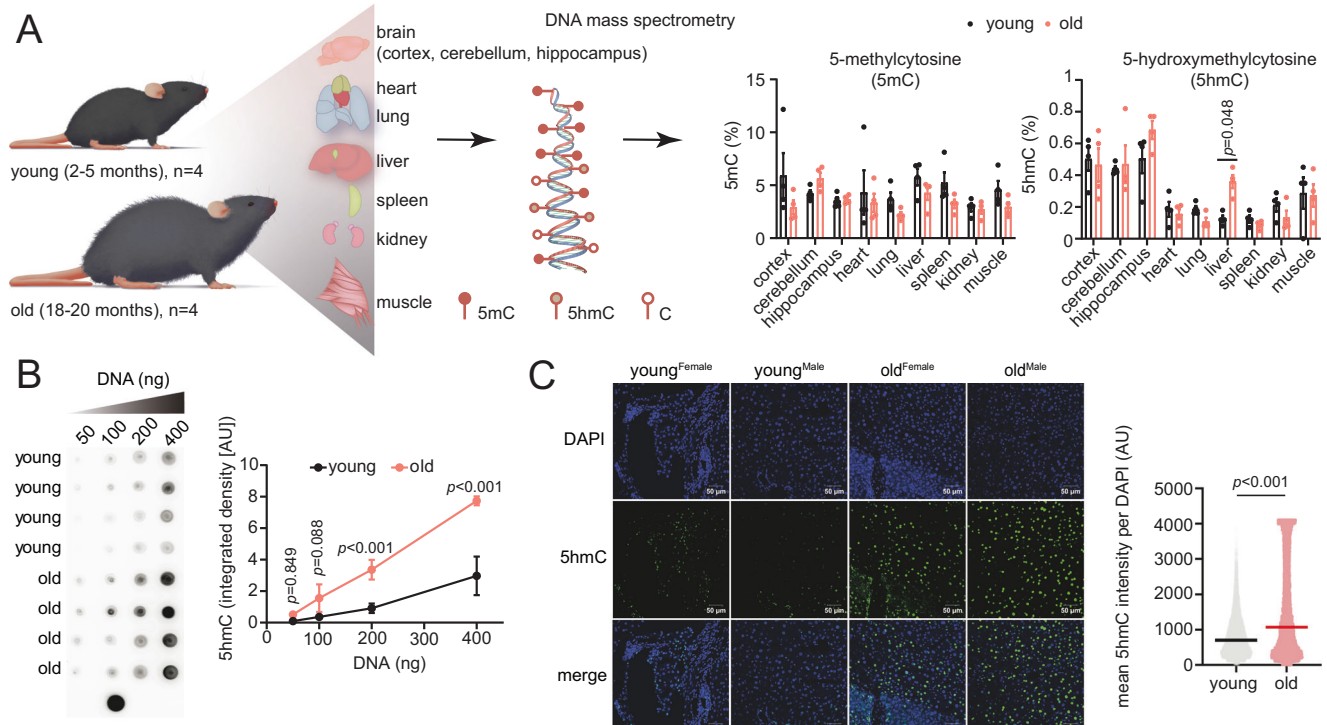

**Fig. 1 | Global accumulation of 5hmC in the aging liver. A** Schematic of experimental procedure and tissues used to profile global levels of 5mC and 5hmC by DNA mass spectrometry (nLC-MS/MS). Data are presented as mean ± SEM; statistical significance was assessed using two-sided unpaired Welch's *t*-test with Holm-Šídák correction for multiple comparisons. **B** Dot blot for 5hmC using increasing amounts of gDNA isolated from young and old ($n = 4$ each) mouse livers; + control is 200 ng of young mouse hippocampus gDNA, − control is water. Signal quantifications are shown on the right. Data are presented as mean ± SD; statistical significance was assessed using two-way ANOVA with Šídák correction for multiple comparisons. AU represents arbitrary fluorescence units. **C** Representative immunofluorescence microscopy for 5hmC in young and old ($n = 2$ each) sex-matched liver sections. The mean 5hmC signal intensity per nucleus is quantified on the right, using data from 10 fields of view for each of the two young and two old biological replicates. Horizontal bar represents median; statistical significance was assessed using two-sided unpaired Welch's *t*-test. Source data are provided as a Source Data file. Illustration credit: Endosymbiont GmbH.

necessarily preclude a reduction of 5mC at local sites (as we show below).

We corroborated the age-related increase of 5hmC in the liver using two orthogonal approaches. Using an antibody against 5hmC and gDNA derived from mouse liver tissue, we detected significantly higher 5hmC signal in the old liver by dot blot analysis (Fig. 1B). Similarly, immunofluorescence microscopy showed significantly higher 5hmC signal in the nuclei of old liver tissue sections compared to young (Fig. 1C). This increase of 5hmC with age was observed for both sexes. Taken together, these data indicate a global increase of 5hmC in the aged liver.

## 5hmC accumulates at genic regions associated with hepatic metabolism during aging

We next investigated the genomic localization of 5hmC using hydroxymethylated DNA immunoprecipitation followed by sequencing (hMeDIP-seq). The hMeDIP-seq was performed on gDNA isolated from the liver of young (~3–4 months) and old (~20–22 months) C57BL/6JN mice by immunoprecipitating (IP) with an antibody targeting 5hmC ($n = 4$ biological replicates per age group, Supplementary Fig. 1A, Supplementary Dataset 1). 10% of gDNA was kept as input and did not go through IP. IP success was verified by qPCR analysis of exogenous hydroxymethylated and unmodified DNA spike-in controls that were added before samples underwent IP (Supplementary Fig. 1B). The sequencing reads were aligned to the GRCm38/mm10 genome assembly and were not significantly different between young and old IP or input samples in terms of sequencing depth, mapped fragments, fragment length, or duplication rate (Supplementary Fig. 1C–G).

To identify the primary source of variance in genome-wide 5hmC signal, we performed principal component analysis (PCA) using the RPKM (reads per kilobase per million) normalized and input subtracted 5hmC samples. Principal component 1 (PC1) accounted for 79.4% of the variation in 5hmC signal and clearly distinguished the samples by age, designating age as a main contributor to variability in genome-wide 5hmC signal (Fig. 2A). To identify age-related differentially hydroxymethylated regions (DHMRs), we used QSEA (quantitative sequencing enrichment analysis)[40]. QSEA identified 16,315 regions with significantly higher 5hmC enrichment in the old (hyper DHMRs, fold change [FC] ≥ 2, FDR < 0.05) and 13,592 regions with significantly higher enrichment in young samples (hypo DHMRs, FC ≤ −2, FDR < 0.05) (Fig. 2B, Supplementary Dataset 3). 5hmC signal at the center of the hyper DHMRs showed reproducibly higher signal for old compared to young, and vice versa for hypo DHMRs (Fig. 2C). Figure 2D shows genome browser example views of hyper and hypo DHMRs.

To gain insight into the functional pathways associated with the DHMRs, we performed Gene Ontology (GO) analysis using the Genomic Regions Enrichment of Annotations Tool (GREAT)[41]. The top pathways enriched for hypo and hyper DHMRs largely constituted metabolic and mitochondrial-related terms, including fatty acid, small molecule, and carboxylic acid metabolic processes (Fig. 2E). The similarity in the GO terms between age-related hypo and hyper DHMRs indicates that 5hmC undergoes dynamic changes at genomic regions associated with metabolic and mitochondrial function. It is noteworthy that the liver is a key regulator of whole-body metabolism, and that 5hmC has been previously reported to mark tissue-specific genes[30,31,35], suggesting that age-related differences in 5hmC may, in general, occur at tissue-specific genes. Consistent with this notion, we assessed transcription factors (TFs) associated with the top 500 genes marked by 5hmC in young and old using Lisa (epigenetic Landscape In-Silico deletion Analysis)[42], which revealed motifs for several TFs that have been previously implicated in liver-specific function, including peroxisome proliferator-activated receptors (PPARγ)[43], T-box transcription factor 3 (TBX3)[44,45], and GLIS Family Zinc Finger 2 (GLIS2)[46] (Supplementary Fig. 1H). We then annotated the DHMRs to CpG and genic features, which revealed that most changes were occurring at

regions farthest (>4 kb) from CpG islands (interCGIs) and in gene bodies, primarily intronic regions (Fig. 2F). Furthermore, when we traced 5hmC signal from the transcription start site (TSS) to the transcription end site (TES) of all known mm10 genes, we observed a pronounced and significant accumulation of 5hmC at gene bodies in the aged liver (Fig. 2G).

Collectively, these analyses show that 5hmC accumulates in the aged liver genome-wide and undergoes dynamic changes at gene bodies and regions associated with liver-specific function, primarily hepatic metabolism.

## Gene body 5hmC restricts the magnitude of transcriptional changes during aging

The age-related accumulation of 5hmC at gene bodies and metabolism-related loci in the liver prompted us to investigate the functional role of gene body 5hmC. We analyzed RNA-seq data that we previously generated from the livers of young (~2 months) and old (~18 months, $n = 3$ biological replicates per age group) mice of both sexes[47] (Supplementary Dataset 1). We first assessed the association between gene body 5hmC and gene expression "*within* each age group" by tracing 5hmC signal over the gene body of protein-coding genes with low, intermediate, and high expression, ranked according to the averaged DESeq2 normalized mRNA count (Supplementary Dataset 4). Consistent with previous studies[30–33,35], we observed a positive association between gene body 5hmC and gene expression for both young (Fig. 3A, left) and old (Fig. 3A, right) age groups.

We next interrogated the role of gene body 5hmC in transcriptional changes "*across* age groups". Spearman's rank-order correlation revealed a significant, albeit weak, negative correlation between young and old mean gene body 5hmC signal and mRNA FC (old vs young) (Supplementary Fig. 1I, panels 1 and 2, note the negative rho). Similarly, we also identified a significant, but weak, negative correlation between 5hmC FC (old vs young) and mRNA FC (Supplementary Fig. 1I, panel 3, note the negative rho). These correlational analyses suggest a relationship between increased 5hmC levels and lower transcriptional changes with age, which we investigated further.

Interestingly, we noted that, in the scatter plots (Supplementary Fig. 1I, panels 1 and 2), genes with lower mean 5hmC signal are broadly dispersed along the $y$-axis (mRNA FC) and as 5hmC increases, the dispersion in mRNA FC decreases and the values center around $y = 0$ (no difference in mRNA between old and young). We confirmed that this distribution was dependent on 5hmC signal by generating random 5hmC values for each gene and correlating them with corresponding mRNA FC. As shown in Supplementary Fig. 1I (panel 4), this correlation was not statistically significant and did not mimic the trend in dispersion observed between young and old 5hmC levels and mRNA FC. To validate this observation, we ranked all detectable protein-coding genes into 100 groups (~169–192 genes per group) based on increasing mean gene body 5hmC signal. We computed the variance (measure of dispersion) in mRNA FC for all 100 groups. As speculated, there was a strong negative correlation between increasing 5hmC signal and variance in mRNA FC for both young and old age groups (Fig. 3B). These observations led us to speculate that 5hmC may be associated with lower dispersion and magnitude of transcriptional changes with age.

To illustrate the negative correlation between 5hmC levels and age-related transcriptional changes, we compared gene expression "*across* age groups". Initially, we ranked and categorized all detectable protein-coding genes based on FC between old vs young (Fig. 3C, Supplementary Dataset 4). This categorization delineates genes *downregulated* or *upregulated* with age. We plotted the 5hmC signal over the gene body of the bottom 33% (downregulated genes, $n = 6340$) and top 33% (upregulated genes, $n = 6339$). Similar to our correlational analyses in Supplementary Fig. 1I, we found that genes downregulated with age had pronounced and significantly higher 5hmC in the old (Fig. 3D, left), whereas genes upregulated with age had

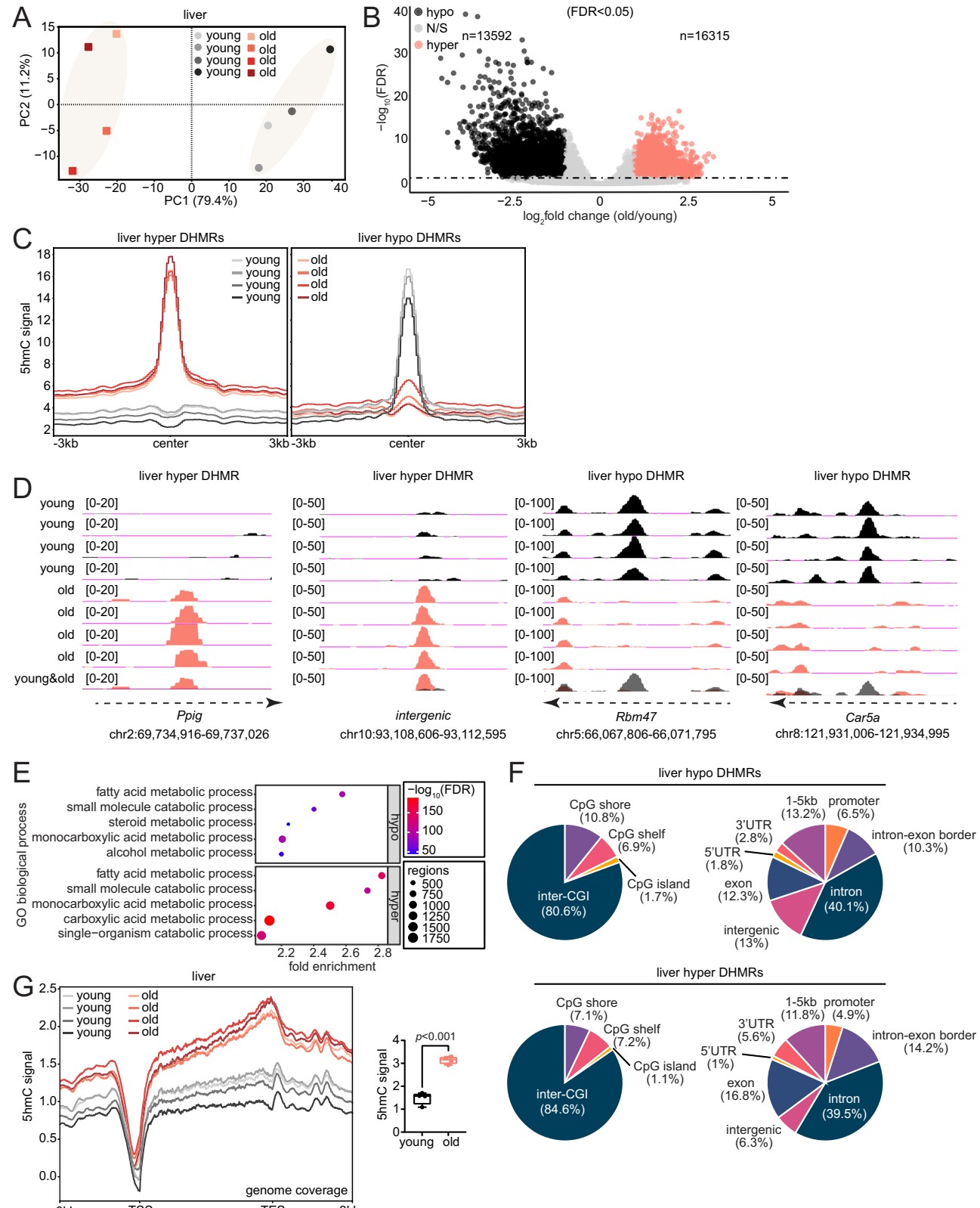

an overall low 5hmC profile (Fig. 3D, right) but still experienced a significant increase in 5hmC with age, albeit with a much smaller effect size.

To assess the relationship between 5hmC and magnitude of transcriptional change, we next ranked and categorized all detectable and protein-coding genes based on absolute FC between old and young, neglecting direction of change (|old vs young|, Fig. 3E,

Supplementary Dataset 4). This categorization delineates genes with *minimal* or *maximal* expression changes with age. We observed that genes with minimal expression change with age (bottom 33%, $n = 6340$) had high gene body 5hmC signal in young samples, which significantly increased in the old (Fig. 3F, left). Conversely, genes with maximal expression change with age (top 33%, $n = 6339$) generally had lower levels of 5hmC (Fig. 3F, right). Given that lowly expressed genes

**Fig. 2 | 5hmC accumulates at genic regions associated with hepatic metabolism during aging. A** Principal component analysis (PCA) plot using input subtracted 5hmC bigWig files of young and old ($n = 4$ each) mice liver. **B** Volcano plot of differentially hydroxymethylated regions (DHMRs) between old and young ($n = 4$ each) mouse liver; identified by QSEA with an FDR < 0.05. Hypo DHMRs (FC ≤ −2) are regions with less enrichment in the old and hyper DHMRs (FC ≥ 2) are regions with higher enrichment in the old. **C** Metaplots of young and old ($n = 4$ each) mouse liver 5hmC signal at the DHMRs identified by QSEA. **D** Example genome browser tracks for mouse liver hyper DHMRs (*Ppig* and an intergenic region) and hypo DHMRs (*Rbm47* and *Car5a*). **E** Gene ontology (GO) terms associated with the DHMRs from (**B**) using GREAT. The top 5 biological process terms with FDR < 0.05

are shown. **F** Pie charts showing CpG and genic/intergenic annotations of the DHMRs from (**B**). **G** Metaplots of young and old ($n = 4$ each) mouse liver 5hmC signal over the gene bodies of all mm10 genes; signal quantifications are shown on the side. Statistical significance was assessed using two-sided unpaired Welch's *t*-test. For the box plot, the horizontal line within each box represents the 50th, while the bounds of the box depict the 25th and 75th percentile of the data. The whiskers extend to the minima (the smallest value within 1.5 times the interquartile range (IQR) below the first quartile, excluding outliers) and the maxima (the largest value within 1.5 times the IQR above the third quartile, excluding outliers). Source data are provided as a Source Data file.

typically exhibit lower levels of 5hmC and are susceptible to noise due to the detection limitations of short-read RNA sequencing, substantial transcriptional changes observed in these genes may partly stem from this inherent variability. Consequently, this could potentially contribute to an amplified change in mRNA expression levels with age. To address this point, we manipulated 5hmC levels in Fig. 6 and assessed the magnitude of transcriptional changes among genes with minimal expression changes.

In summary, these data suggest a dual role for 5hmC in (1) restricting the magnitude of gene expression changes with age, potentially preserving homeostatic expression of tissue-specific genes, and (2) promoting downregulation of transcription which may decrease tissue-specific functions with age. Notably, in model organisms lacking 5hmC, such as *Caenorhabditis elegans* and *Drosophila melanogaster*, another gene body-associated epigenetic modification, H3K36me3, has been implicated in age-related transcriptional restriction. In that case, a global reduction in H3K36me3 leads to higher variability in age-dependent transcriptional changes and shortened lifespan[48]. Similarly, broad H3K4me3 domains, marking cell identity genes, are linked to "transcriptional consistency", akin to transcriptional restriction, across various human and mouse cells[49].

## Age-related differences in 5hmC occur without detectable differences in 5mC

Since 5hmC is a product of 5mC oxidation, we inquired whether an age-related transcriptionally restrictive function for 5hmC would also be apparent for 5mC. Accordingly, we profiled 5mC genome-wide in gDNA isolated from liver tissue of the same mice as the hMeDIP-seq using methylated DNA immunoprecipitation followed by sequencing (MeDIP-seq) with an antibody targeting 5mC (QC metrics in Supplementary Fig. 2A-G, the MeDIP data is reported in Yang et al.[47]). In PCA, PC1 accounted for 31.8% of the variability in genome-wide 5mC signal and showed a modest clustering of the samples by age (Supplementary Fig. 3A). Differential analysis of old vs young using the same QSEA parameters and threshold as the liver hMeDIP-seq resulted in no genomic window surviving FDR < 0.05. To enable downstream comparison with the liver hMeDIP-seq data, we relaxed the threshold for statistical significance and classified differentially methylated regions (DMRs) using $p < 0.05$. Using these parameters, we obtained 42,488 total DMRs, 21,148 with greater enrichment in old (FC ≥ 2, $p < 0.05$) and 21,340 with greater enrichment in young (FC ≤ −2, $p < 0.05$) (Supplementary Fig. 3B, Supplementary Dataset 3). We emphasize that relaxing the statistical threshold may lead to some false positive differential enrichments, and thus, overall 5mC differences between young and old are modest compared to 5hmC. Nevertheless, verification of the 5mC signal at the center of the hyper DMRs showed reproducibly higher signal for old compared to young, and vice versa for the hypo DMRs (Supplementary Fig. 3C, left). Moreover, 5mC signal at the liver *DHMRs* (from Fig. 2B) showed that hyper DHMRs had overall higher 5mC signal than hypo DHMRs, indicating that 5mC to 5hmC conversion likely contributed to the higher 5hmC enrichment with age (Supplementary Fig. 3C, right). Example genome browser views of

hyper and hypo DMRs are shown for individual replicates in Supplementary Fig. 3D. In contrast to the liver DHMRs which were prominently enriched for metabolic terms (Fig. 2E), GO analysis of the liver DMRs using GREAT revealed terms related to endoplasmic reticulum stress and neuron differentiation (Supplementary Fig. 3E). Given that previous work comparing 5mC and 5hmC has shown that 5hmC is a better marker of tissue-specific genes than 5mC[31], it is reasonably expected that the liver DHMRs would be mostly associated with liver-specific function (i.e., metabolic function) as opposed to the DMRs. CpG and genic annotations of the DMRs were comparable with the DHMRs, wherein most DMRs were associated with interCGIs and gene bodies, primarily intronic regions (Supplementary Fig. 3F). 5mC signal at gene bodies of all mm10 genes did not show significant differences in 5mC with age (Supplementary Fig. 3G) in contrast to 5hmC (Fig. 2G). Overall, these results confirm previous reports that 5hmC is a better marker of tissue-specific genes than 5mC[31] and show that age-related differences in 5hmC occur without pronounced detectable differences in 5mC at these locations.

We next assessed whether 5mC also predicted the magnitude of transcriptional changes with age by tracing 5mC signal at gene bodies that show minimal and maximal changes with age (Fig. 3E). In contrast to what we observed for 5hmC patterns (Fig. 3F), we found no significant differences in 5mC signal for either group of genes (Fig. 3G). Genes that underwent minimal expression changes with age (Fig. 3G, left), however did trend towards lower 5mC signal in the old, in agreement with the conversion to 5hmC in the old.

In general, the genes that underwent dramatic transcriptional changes with age and displayed lower 5hmC signal (Fig. 3F, right), tended to have significantly shorter 5' UTRs (untranslated regions), 3' UTRs, transcript length, CDS (coding sequence) length, and had fewer number of exons compared to the genes that were transcriptionally restricted with age (Fig. 3H). This suggests that relatively longer genes may be more prone to 5hmC accumulation.

Overall, our results show that genes that undergo minimal changes in expression with age are characterized by a pronounced and significant accumulation of gene body 5hmC during aging but with a modest and non-statistically significant decrease of 5mC. This suggests that although 5hmC is catalyzed from 5mC, gene body 5hmC levels are stably enriched in aged tissues and may possibly exert a greater influence on age-related transcriptional changes.

## Alternative splicing mediates 5hmC's transcriptionally restrictive function through decreased binding of splicing factors

DNA modifications have been shown to influence transcriptional activity via recruitment of proteins that alter chromatin architecture or transcription factor binding[50]. Accordingly, we sought to identify 5hmC-protein interactors as a potential mechanism through which the modification might restrict the magnitude of transcriptional changes with age. We performed oligonucleotide mass spectrometry using nuclear extracts, prepared from young (~5 months) and old (~ 20 months, $n = 4$ biological replicates per age group) mouse liver tissue from both sexes, and three 20 bp DNA oligos that were either unmodified (C), methylated (5mC), or hydroxymethylated (5hmC) at

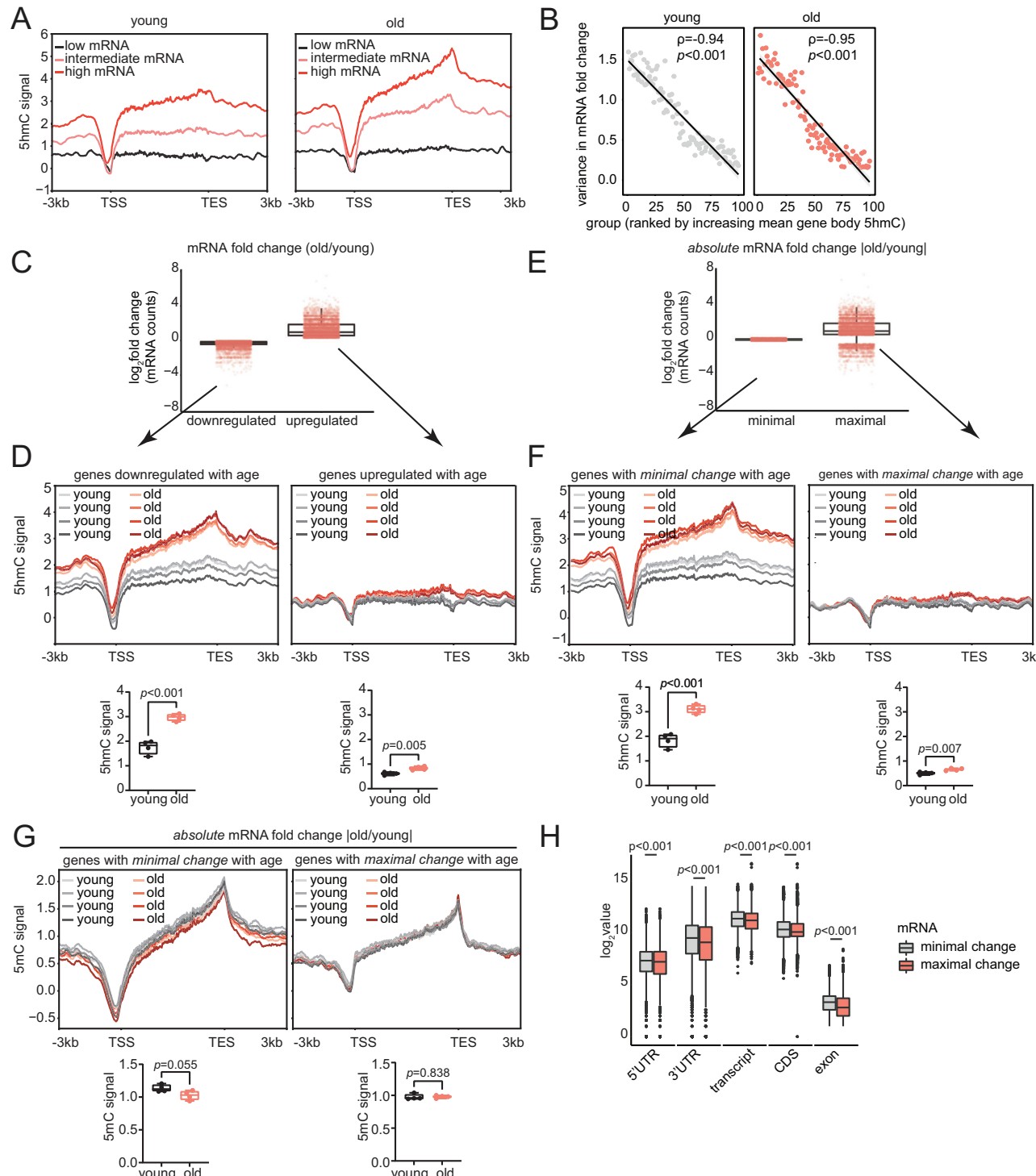

**Fig. 3 | Gene body 5hmC restricts the magnitude of transcriptional changes during aging. A** Metaplots of merged young and old ($n = 4$ each) mouse liver 5hmC signal over gene bodies with low ($n = 6,340$), intermediate ($n = 6,339$), and high ($n = 6,339$) average mRNA counts for young (left) and old (right) samples ($n = 3$ each). **B** Correlation between average gene body 5hmC signal (100 ranked groups) and variance in mRNA FC (old vs young) among the genes per group for young (left) and old (right). ρ = Spearman's correlation coefficient, $p$-values were derived from Spearman's rank correlation. **C** Box plots showing mRNA FC of old vs young ($n = 3$ each) for genes downregulated or upregulated with age. **D** Metaplots of young and old ($n = 4$ each) mouse liver 5hmC signal over gene bodies in (**C**). Quantifications are depicted below the plot; statistical significance was assessed using two-sided unpaired Welch's $t$-test. **E** Same as (**C**) except for genes with minimal or maximal expression change between old and young ($n = 3$ each). **F** Metaplots of young and old ($n = 4$ each) mouse liver 5hmC signal over gene bodies in (**E**). Quantifications are

depicted below the plot; statistical significance was assessed using two-sided unpaired Welch's $t$-test. **G** Metaplots of young and old ($n = 4$ each) mouse liver 5mC signal over gene bodies in (**E**) with minimal (left) and maximal (right) expression change with age. Quantifications are depicted below the plot; statistical significance was assessed using two-sided unpaired Welch's $t$-test. **H** Box plots showing the distribution of various genic features for the genes with minimal and maximal expression changes between old vs young ($n = 3$ each) mice. Statistical significance was assessed using two-sided unpaired Welch's $t$-test. For all box plots (**C**–**H**), the horizontal line within each box represents the 50th, while the bounds of the box depict the 25th and 75th percentile of the data. The whiskers extend to the minima (the smallest value within 1.5 times the IQR below the first quartile, excluding outliers) and the maxima (the largest value within 1.5 times the IQR above the third quartile, excluding outliers). Source data are provided as a Source Data file.

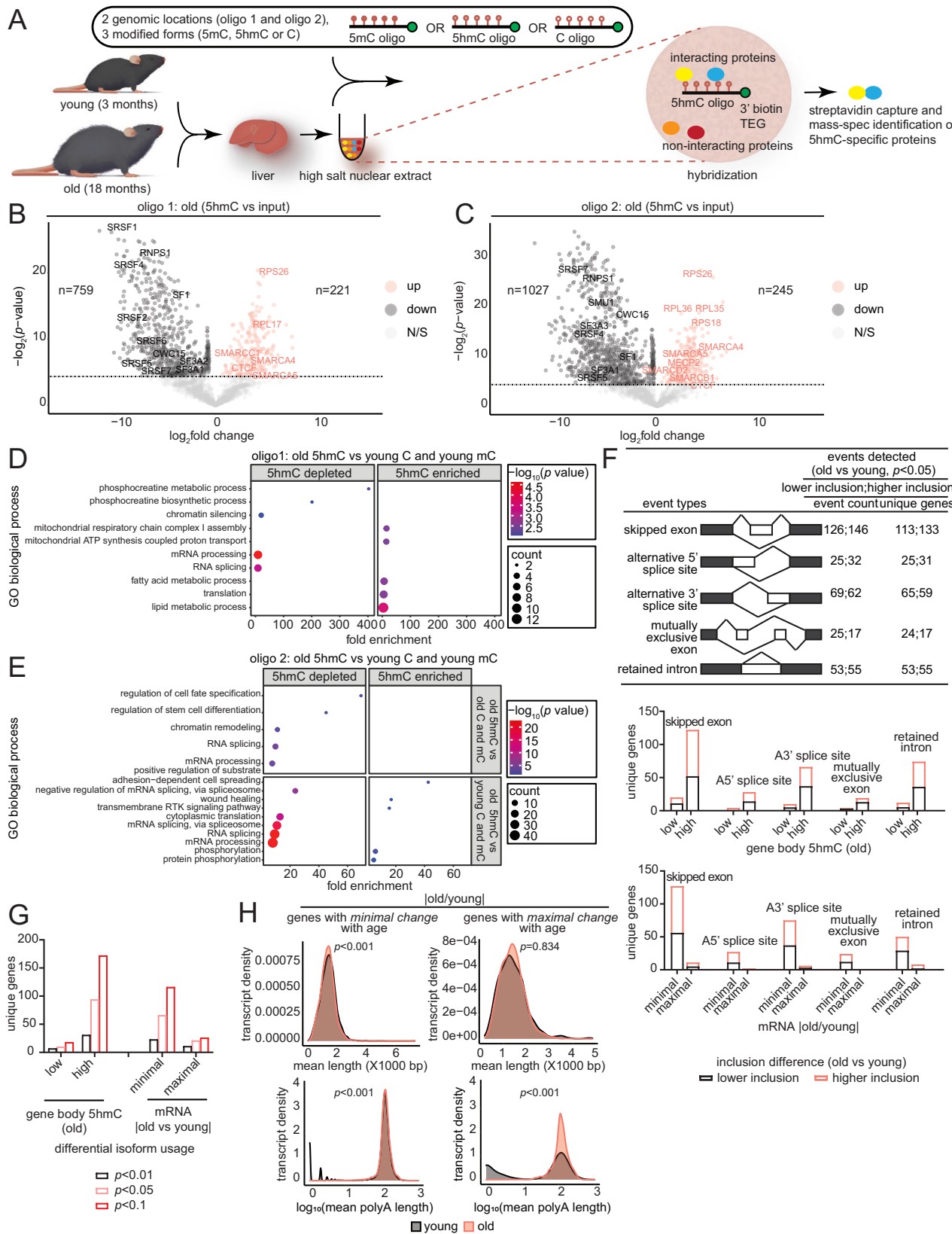

the cytosine nucleotides (Fig. 4A, Supplementary Dataset 1 and 5). The oligos were designed using an endogenous DNA sequence within two hyper DHMRs (Fig. 2B) resulting in 48 pull-down assays with 6 oligos (2 unique regions with 3 modifications) and 8 mouse liver extracts.

Genome browser views of the regions used to design the oligos are shown in Supplementary Fig. 4A. Following streptavidin capture, bound peptides from the DNA-pull downs as well as 10% input were identified by mass spectrometry.

**Fig. 4 | Alternative splicing mediates 5hmC's transcriptionally restrictive function through decreased binding of splicing factors. A** Schematic of oligo mass-spec experimental procedure. **B** Volcano plot showing differentially enriched proteins in old mice for the 5hmC oligo 1 pull-down vs input ($n = 4$ each). Some significantly enriched or de-enriched interactors are labeled. **C** Same as (**B**) except for the 5hmC oligo 2 pull-down vs input ($n = 4$ each) in the old. **D** GO terms associated with proteins depleted/de-enriched or enriched for oligo 1 in the old 5hmC vs young C and mC ($n = 4$ each) comparison. **E** GO terms associated with proteins depleted/de-enriched or enriched for oligo 2 in the old 5hmC vs old C and mC ($n = 4$ each) comparison (top) and the old 5hmC vs young C and mC ($n = 4$ each) comparison (bottom). **F** Number of differential splicing events detected in RNA-seq data between old and young ($n = 3$ each) samples at $p < 0.05$ using rMATS (top). Number of events and the unique number of genes are indicated. Number of

differential splicing events grouped by increasing gene body 5hmC signal in old (middle). Number of differential splicing events grouped by minimal or maximal expression change with age (bottom). **G** Bar plots showing differential isoform usage from dRNA-seq results with young and old ($n = 4$ each) samples for genes with minimal and maximal expression changes with age at indicated $p$-value thresholds derived from the rMATS statistical model (**H**) Transcript length (top) and poly A length (bottom) distribution for genes that undergo minimal or maximal expression changes between old and young ($n = 4$ each); statistical significance was assessed using Mann–Whitney $U$ test. For (B-E), statistical differences for each protein were assessed using Welch's $t$-test (if the $F$-test $p$-value was <0.05); otherwise, the standard Student's $t$-test was used. Source data are provided as a Source Data file. Illustration credit: Endosymbiont GmbH.

We identified 221 proteins in the nuclear extract from the old mouse liver samples that were significantly enriched in the oligo 1 5hmC pull-down and 759 that were depleted (5hmC vs input) (Fig. 4B, Supplementary Dataset 6). For oligo 2, 245 proteins were enriched in the 5hmC pull-down and 1027 were depleted (Fig. 4C). 49% (154) of the proteins were found to be commonly enriched in both oligo 1 and oligo 2 5hmC pull-downs (Supplementary Fig. 4B, top), while 53% (619) were found to be commonly depleted (Supplementary Fig. 4B, bottom). The common 5hmC-enriched proteins were associated with biological processes relating to translation, transcription, and chromatin regulation, while the commonly depleted proteins were associated with metabolic processes and RNA splicing (Supplementary Fig. 4C).

We next assessed the specificity of protein interactions for 5hmC, by measuring differences in abundance of these proteins in 5hmC, 5mC, or C pull-downs in the old, as a potential mechanism underlying the relationship between 5hmC and transcriptional restriction or propensity for downregulation. For oligo 1, we identified 3 proteins that were significantly depleted for the 5hmC modification compared to 5mC and C, which included the NuA4 histone acetyltransferase complex associated protein, MORF4L2 (Supplementary Fig. 4D, in red). 12 proteins were significantly enriched for 5hmC, two of which are known to be involved in the endoplasmic reticulum stress response (HYOU1[51] and TMEM259[52], Supplementary Fig. 4D, in red), a condition previously linked to changes in chromatin architecture[53]. By contrast, we found 90 proteins that were significantly enriched in the 5hmC oligo 1 pull-down and 49 that were depleted compared to 5mC and C oligos in the young (Supplementary Fig. 4E). GO terms for the proteins depleted in the 5hmC pull-downs in the old, included RNA splicing, mRNA processing, and chromatin silencing, while the proteins enriched in the 5hmC pull-down were associated with translation, metabolic and mitochondrial processes (Fig. 4D).

For oligo 2, we identified 3 proteins (PISD, G3BP1, PA2G4) that were significantly enriched in the 5hmC modification compared to 5mC and C in the old (Supplementary Fig. 4F). Notably, G3BP1 is a DNA and RNA-binding protein implicated in mRNA decay[54] and previously reported to regulate steady state levels of mRNAs with highly structured 3'UTRs[55]. Thus, 5hmC's binding to G3BP1 would agree with the correlation observed between 5hmC and age-related transcriptional restriction (Fig. 3E, F) as well as the modest repressive correlation observed in Fig. 3C, D. Moreover, GO analysis of the proteins depleted for 5hmC for oligo 2 were similar to oligo 1 in showing enrichment of GO terms associated with RNA splicing, mRNA processing, and chromatin regulation (Fig. 4D, E). Additionally, comparisons between 5hmC oligo in the old and 5mC and C in the young revealed that proteins depleted for 5hmC were involved in mRNA processing and multiple RNA splicing-related GO terms (Supplementary Fig. 4G and Fig. 4E, bottom). Altogether, these results show that 5hmC consistently has decreased affinity for proteins involved in splicing compared to 5mC and C within aged tissues and during aging.

The prominent depletion of RNA splicing-related proteins in the aged liver for the majority of the 5hmC oligo pull-downs, coupled with

the pronounced age-related changes in 5hmC at intronic regions (Fig. 2F) motivated us to investigate alternative splicing during aging. Using rMATS (Multivariate Analysis of Transcript Splicing)[56], we detected several alternative splicing events between old and young liver samples in bulk RNA-seq data (Fig. 4F, top, Supplementary Dataset 7). We found that genes with relatively higher gene body 5hmC in the old experienced the most alternative splicing events with age (Fig. 4F, middle). In agreement, genes that underwent minimal expression changes with age also experienced the most age-related alternative splicing events (Fig. 4F, bottom).

Although isolated alternative splicing events can be detected with short-read sequencing (for example with rMATS), comprehensive analysis of splice isoforms is limited. With nanopore sequencing, where read length is equal to fragment length, entire transcripts can be sequenced in single reads giving a more detailed view of isoform diversity. We thus performed direct RNA-seq (dRNA-seq) with nanopore sequencing (Oxford Nanopore Technologies) on the MinION platform with livers from young (~2 months) and old (~18 months, $n = 4$ biological replicates per age group) C57BL/6JN mice of both sexes (Supplementary Dataset 1). dRNA-seq is free from PCR bias and can provide information on expression, splicing isoforms, transcript length and poly A length. PCA plots of normalized transcript counts clearly segregated the samples by sex in PC1 and age in PC2 (Supplementary Fig. 4H). A comparison of the differential isoform usage from long-read data (Supplementary Dataset 8) showed similar trends to rMATS results (Fig. 4F), thus providing additional validation of increased alternative splicing events in genes marked by high 5hmC in old, i.e., genes showing minimal change with age (Fig. 4G). A few examples of these alternative splicing events are shown in Supplementary Fig. 4I. A survey of the mean transcript lengths and poly A lengths in these genes showed a small but significant decrease in overall transcript length (mean difference = −45.5 bp) but slight increase in poly A length (mean difference = 13.1) in the old (Fig. 4H, top left and bottom left, Supplementary Dataset 9). The length difference could indicate a prevalence of short isoforms or ongoing mRNA decay in old tissue. By contrast, genes undergoing maximal changes in gene expression with age showed no evidence of transcript length change (Fig. 4H, top right plot, Supplementary Dataset 9). The poly A length of this set of genes however, showed a prominent increase (mean difference = 48.5) in the old, suggesting that the corresponding transcripts are relatively stabilized. Several studies have shown that splicing factors can influence transcriptional elongation rates by modulating pol II elongation, and vice versa[57–59]. Splicing factors have also been shown to regulate steady state mRNA levels by producing alternatively spliced transcripts targeted for degradation by the nonsense-mediated decay pathway[60–63]. Together, these events could help fine-tune homeostatic expression of tissue-specific genes. Thus, our data suggest a potential role for RNA splicing in mediating 5hmC's age-related transcriptionally restrictive function.

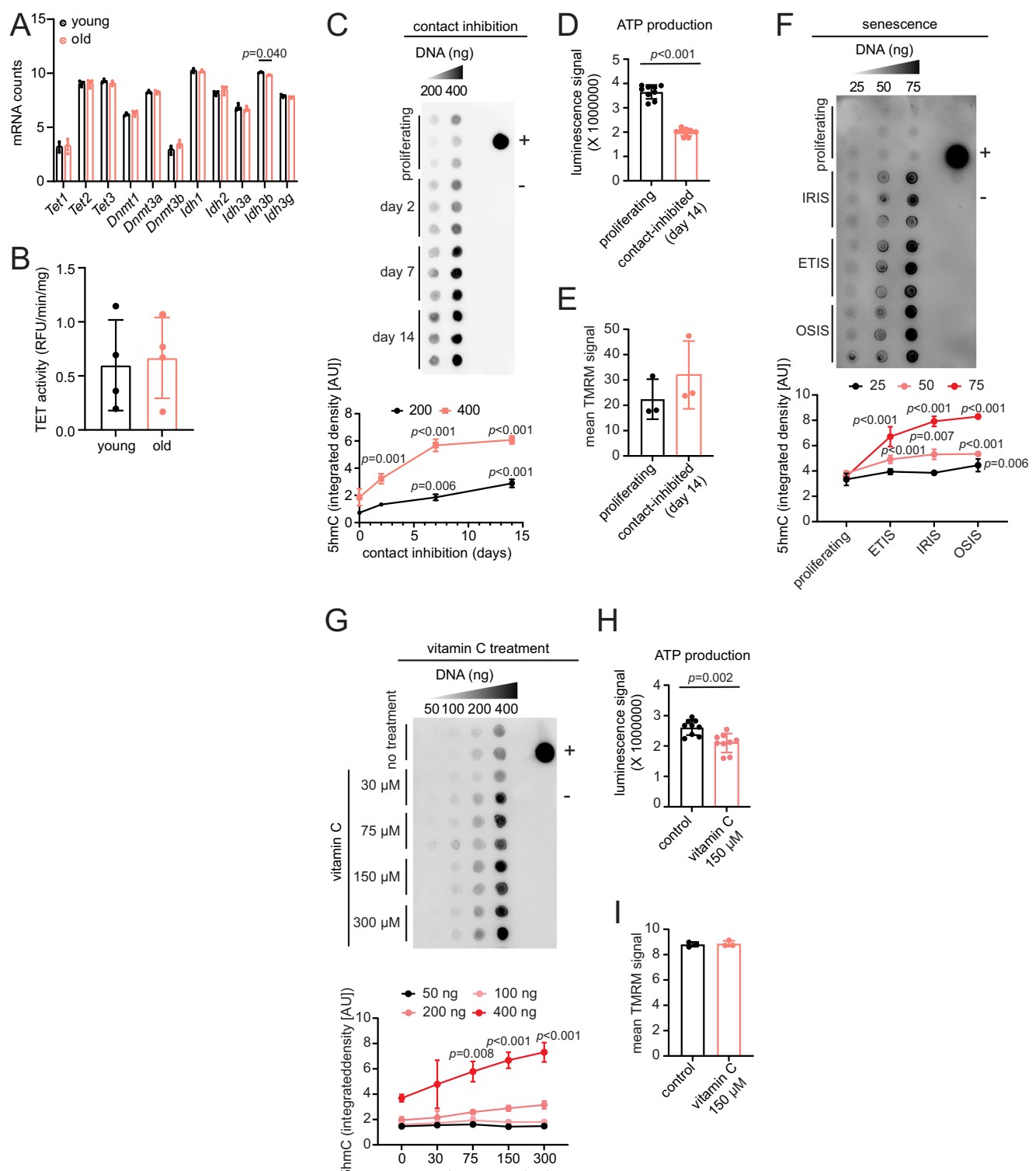

## Quiescence and senescence drive the increase of 5hmC with age and impact ATP production

We next sought to address the mechanisms responsible for the increase of 5hmC in the aged liver. RNA-seq data showed no significant differences in mRNA levels of 5hmC regulators, i.e., *Tet1*, *Tet2*, and *Tet3*, between old and young (Fig. 5A). We did, however, detect a significant age-related decrease in a gene with indirect links to 5hmC, *Idh3b* (isocitrate dehydrogenase 3, beta subunit). However, because isocitrate dehydrogenases are responsible for converting isocitrate to α-ketoglutarate, a cofactor for TET enzymes, their age-related decrease

in expression contradicts the increase of 5hmC in the aged liver. We further investigated whether TET activity was altered in the aged liver and found no significant differences (Fig. 5B). These results indicate that the increase of 5hmC with age is not a simple consequence of cognate enzyme abundances or activity.

Studies have shown that TET proteins have higher enzymatic activity towards 5mC than 5hmC, and therefore are more prone to write than erase 5hmC from the genome[23]. By contrast, DNA replication has been reported to dilute 5hmC[20,21]. This led us to hypothesize that in the aged liver, where cells are not actively dividing and are in

**Fig. 5 | Quiescence and senescence drive the increase of 5hmC with age and impact ATP production. A** Normalized mRNA counts in young and old ($n = 3$ each) mouse liver for 5hmC-relevant enzymes. Data are presented as mean ± SEM; statistical significance was assessed using multiple two-sided unpaired $t$-test with FDR correction (Benjamini, Krieger, and Yekutieli). **B** TET activity assay in young and old ($n = 4$ each) mouse liver lysates. **C** Dot blot for 5hmC signal using gDNA isolated from proliferating and contact inhibition-induced quiescent HepG2 cells ($n = 3$ independent cell cultures sourced from the same vial). + control is 200 ng of young mouse hippocampus gDNA, – control is water. Quantifications are depicted below. **D** ATP production assay using proliferating and contact-inhibited quiescent HepG2 cells ($n = 3$ technical replicates for each of 3 independent cell cultures sourced from the same vial). **E** TMRM mitoprobe assay with proliferating and contact-inhibited quiescent HepG2 cells ($n = 3$ independent cell cultures sourced from the same vial). **F** Dot blot of 5hmC using gDNA isolated from proliferating and IRIS, ETIS, and OSIS WI-38 cells ($n = 3$ independent cell cultures sourced from a single vial). +control is

50 ng of young mouse hippocampus gDNA, – control is water. Quantifications are shown below. **G** Dot blot for 5hmC using gDNA isolated from HepG2 cells treated with vitamin C ($n = 2$ independent cell cultures sourced from a single vial). + control is 200 ng of young mouse hippocampus gDNA, – control is water. Quantifications are depicted below. **H** ATP production assay using proliferating and vitamin C treated HepG2 cells ($n = 3$ technical replicates for each of 3 independent cell cultures sourced from the same vial). **I** TMRM mitoprobe assay with proliferating and vitamin C treated HepG2 cells ($n = 3$ independent cell cultures sourced from the same vial); statistical significance was assessed using two-sided unpaired Welch's $t$-test. For panels (**B**–**I**), data are presented as mean ± SD. Statistical significance was assessed using two-sided unpaired Welch's $t$-test, except for dot blots (**C**, **F**, **G**), which used two-way ANOVA with Tukey's multiple comparisons post-hoc test. AU represents arbitrary fluorescence units. Source data are provided as a Source Data file.

prolonged quiescence in a nutrient-rich environment (a state that we have previously called hyper-quiescence[47]), genomic increases of 5hmC are likely. To test this hypothesis, we induced quiescence in HepG2 cell lines in two ways: by contact inhibition, a model that mimics the state of non-dividing organs in two dimensions, and by serum starvation. We detected a significant increase of 5hmC with longer times in quiescence induced by contact inhibition (Fig. 5C) but not serum starvation (Supplementary Fig. 5A), suggesting that the hyper-quiescent state contributes to the accumulation of 5hmC.

Given that 5hmC marks mitochondrial and metabolic genes in the aged liver, coupled with the observed correlation between 5hmC and mRNA downregulation (Fig. 3C, D), we further assessed whether the increase in 5hmC would influence metabolic function. We increased 5hmC by contact inhibition-induced quiescence and observed significantly lower total ATP production compared to proliferating cells (Fig. 5D). Notably, there were no significant differences in the mitochondrial membrane potential of the quiescent cells compared to proliferating cells (Fig. 5E), suggesting that the reduced ATP levels with high 5hmC is not due to a loss of mitochondrial membrane integrity.

Having observed that cell cycle arrest can drive the increase of 5hmC, we next inquired whether 5hmC levels might also be higher in senescence, a state of stable cell cycle arrest. Senescent cells are known to accumulate in aged tissues, contributing to age-related decline, and thus may partially contribute to the increase of 5hmC with age. We induced senescence in human WI-38 cells by treatment with etoposide (ETIS), ionizing ($\gamma$) radiation (IRIS), or oxidative stress (OSIS) for 10 days. Cells were confirmed to be in the senescence state by the level of senescence-associated beta-galactosidase (SA-β-gal) staining (Supplementary Fig. 5B), increased expression of known senescence-associated markers *p16*, *p21*, and *Il-6*; downregulation of *Lmnb1* (Supplementary Fig. 5C); and reduced BrdU labeling (Supplementary Fig. 5D). We found that senescent cells, either ETIS, IRIS, or OSIS, had significantly higher levels of genomic 5hmC compared to proliferating controls (Fig. 5F). Our results suggest that the accumulation of senescent cells with age may contribute to the increase of 5hmC in the aged liver.

Previous studies have shown that ROS can promote conversion of 5mC to 5hmC[17]. Consequently, we wondered whether age-related oxidative stress might also contribute to the accumulation of 5hmC in aged liver. Indeed, dihydroethidium (DHE) staining of mouse liver sections revealed significantly higher ROS signal in old liver nuclei compared to young (Supplementary Fig. 5E and F). To directly assess whether ROS influences genomic 5hmC, we induced cellular ROS in HepG2 cells using an acute 2 h treatment with 600 μM hydrogen peroxide ($H_2O_2$), which was shown to be sufficient to increase ROS levels (Supplementary Fig. 5G). We also sought to decrease $H_2O_2$-induced ROS by either 24 h sequential or co-treatment with a radical scavenger, N-acetylcysteine amide (NAC), shown to reduce $H_2O_2$-induced ROS at

2 mM concentration (Supplementary Fig. 5H). We found that acute $H_2O_2$ treatment did not significantly increase 5hmC signal, and reducing ROS, via sequential or co-treatment with NAC, did not influence genomic 5hmC levels (Supplementary Fig. 5I). Chronic 24 h treatment with $H_2O_2$ also had no effect on 5hmC production (Supplementary Fig. 5J). Thus, in our hands, ROS itself was not sufficient to increase global 5hmC levels.

Since quiescence and senescence can influence metabolic and mitochondrial function independent of changes in 5hmC, we opted to assess the functional effects of 5hmC using vitamin C (ascorbic acid). Vitamin C can drive the increase of 5hmC by acting as a co-factor for the TET proteins and enhancing oxidation of 5mC to 5hmC[18,19]. Indeed, we observed a positive association between vitamin C treatment and 5hmC production (Fig. 5G) in HepG2 cells as reported previously in other models[18,19,64]. Similar to contact inhibition-induced quiescence (Fig. 5C–E), increasing 5hmC by vitamin C led to a significant decrease in ATP production (Fig. 5H) with no significant differences in mitochondrial membrane potential (Fig. 5I). Overall, our data suggest that age-related contexts such as quiescence and senescence, or vitamin C treatment, but not ROS, can increase 5hmC and downregulate tissue-specific function.

## Altering 5hmC levels affects transcriptional magnitude

To establish a direct link between 5hmC and lower magnitude of transcriptional changes, we sought to manipulate 5hmC levels and then assess transcriptional outcomes. Given the increase of 5hmC associated with quiescence (Fig. 5C) and previous data on DNA replication-induced reduction of 5hmC levels[20–22], we reasoned that liver regeneration may dilute 5hmC levels. Accordingly, we performed 70% partial hepatectomy in young (~3–4 months) and old (~20–22 months) mice of both sexes ($n = 3$ biological replicates per time point, Fig. 6A) and collected liver samples pre-surgery and 48, 72, 96, 120, and 240 h post-surgery, as reported in Yang et al.[47]. Our pre-surgery and post-surgery livers were derived from the same animals (Supplementary Dataset 1). Results from nLC-MS/MS showed a significant combined effect of age and regeneration on relative global 5hmC levels (Fig. 6B). Importantly, we observed a progressive dilution of the age-accumulated 5hmC with liver regeneration. For subsequent analyses, we used the 240 h post-surgery samples as liver regeneration is deemed complete by that time. Upon regeneration, local 5hmC levels, measured by hMeDIP-seq, were strongly reduced over gene bodies (Fig. 6C) and this effect was particularly remarkable for genes that showed minimal expression changes with age (Fig. 6D, left). Notably, this gene set had high levels of 5hmC in the old pre-surgery samples. Interestingly, the dilution of 5hmC after regeneration significantly increased the magnitude of transcriptional changes between old and young in this gene set, consistent with the relationship observed between 5hmC and transcriptional restriction (Fig. 6E). This increase of transcriptional magnitude upon regeneration was also

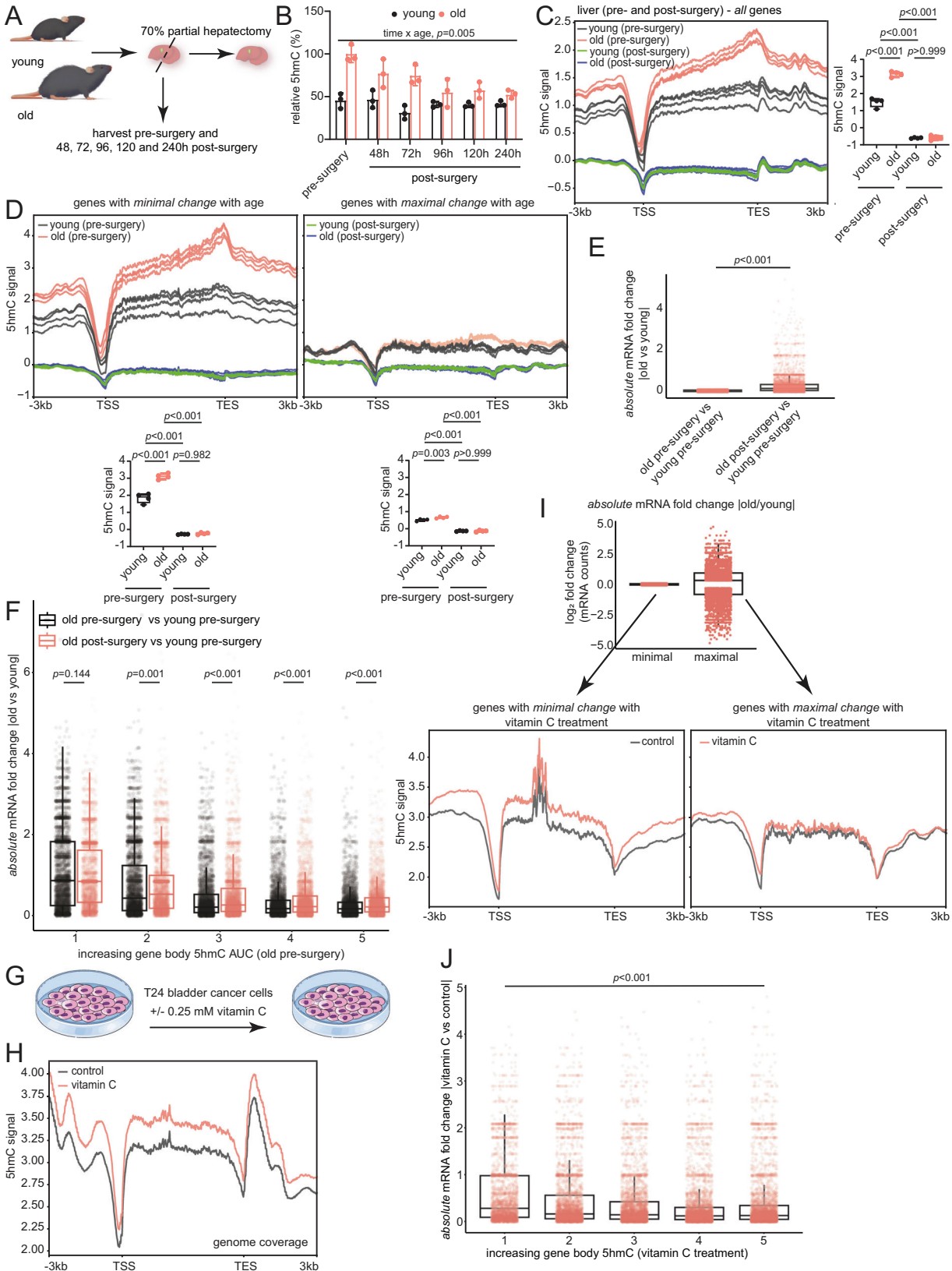

evident when we ordered genes based on increasing gene body 5hmC signal in the old samples (Fig. 6F).

To assess the effects of global increase of 5hmC on transcriptional magnitudes, we mined matched hMeDIP-seq and RNA-seq data from Peng et al.[64], who used untreated and 0.25 mM vitamin C treated human T24 bladder cancer cells (Fig. 6G). We verified that vitamin C

treatment increased 5hmC over bodies of all hg19 genes (Fig. 6H). Similar to the liver (Fig. 3E, F), we ranked genes by absolute mRNA FC between vitamin C and control samples to obtain genes that undergo minimal or maximal expression change with treatment (Fig. 6I, top). We observed that genes with minimal expression change upon vitamin C treatment (bottom 33%, $n = 5605$) had higher gene body 5hmC signal

**Fig. 6 | Altering 5hmC levels affects transcriptional magnitude. A** Schematic for 70% partial hepatectomy. **B** Relative global 5hmC signal for young and old ($n = 3$ each) mouse liver samples at the indicated times. Data are presented as mean ± SD. Statistical significance was assessed using two-way ANOVA with Geisser-Greenhouse correction. **C** Metaplot of 5hmC signal over bodies of all mm10 genes for indicated groups. Quantifications are shown on the side. **D** Metaplot of 5hmC signal across genes bodies with minimal (left) and maximal (right) expression changes between old pre-surgery vs young pre-surgery ($n = 3$ each) mRNA comparisons. Quantifications are shown below. For (**C-D**), statistical significance was assessed using one-way ANOVA with Tukey's multiple comparisons post-hoc test. **E** Box plots showing transcriptional changes for "genes with minimal change with age" (from **D**, left). Statistical significance was assessed using Mann−Whitney *U* test. **F** Box plots showing transcriptional changes for indicated group comparisons. Statistical significance was assessed using a Mann−Whitney U test with FDR correction (Benjamini-Hochberg). **G** Schematic showing vitamin C treatment in T24

bladder cancer cells from Peng et al.[64]. **H** Metaplot of 5hmC signal over the bodies of all hg19 genes in vitamin C-treated T24 cells and untreated controls ($n = 1$ each). **I** Box plots showing absolute mRNA FC distribution of genes with minimal and maximal expression changes in vitamin C-treated T24 cells vs untreated controls ($n = 2$ independent cell cultures)[64]. Below, metaplot of 5hmC signal across gene bodies with minimal (left) and maximal (right) expression changes. **J** Box plots showing transcriptional changes for all genes in vitamin C-treated T24 cells vs untreated controls ($n = 2$ independent cell cultures per group). Statistical significance was assessed using one-way ANOVA. For all box plots, the horizontal line within each box represents the 50th, while the bounds of the box depict the 25th and 75th percentile of the data. The whiskers extend to the minima (the smallest value within 1.5 times the IQR below the first quartile, excluding outliers) and the maxima (the largest value within 1.5 times the IQR above the third quartile, excluding outliers). Source data are provided as a Source Data file. Illustration credit: Endosymbiont GmbH.

in the vitamin C treated cells while those with maximal expression change (top 33%, $n = 5604$) had relatively lower 5hmC levels (Fig. 6I, bottom). Consistent with the link between gene body 5hmC and transcriptional restriction, this relationship was also evident when we ranked genes by increasing gene body 5hmC levels and we observed a gradual and significant decrease in transcriptional magnitude (Fig. 6J).

Altogether, these data show that reducing (liver regeneration) or increasing (vitamin C treatment) 5hmC lead to increases and decreases of transcriptional magnitudes, respectively. These results indicate that 5hmC plays a pivotal role in regulating transcriptional changes with age.

## 5hmC's transcriptionally restrictive function extends to mouse cerebellum

5hmC has been previously reported to be highly abundant in brain tissue[65], which was corroborated by our data (Fig. 1A). Therefore, we wondered whether the transcriptionally restrictive function for 5hmC observed in the liver (Fig. 3E, F) extended to the brain, even though brain regions did not show a significant global increase of 5hmC with age (Fig. 1A). We profiled 5hmC genome-wide using hMeDIP-seq and generated corresponding RNA-seq data for the cerebellum from young (~4−5 months) and old (~21−24 months) C57BL/6JN mice of both sexes ($n = 4$ biological replicates per group, Supplementary Dataset 1, QC metrics in Supplementary Fig. 6A−G). In PCA, PC1 accounted for 64.8% of the variation in genome-wide 5hmC signal and clustered the samples by age (Supplementary Fig. 7A). Differential analysis of old vs young using QSEA resulted in only 74 genomic windows surviving FDR < 0.05. To enable downstream comparison with the liver hMeDIP-seq and MeDIP-seq data, we relaxed the threshold for statistical significance and classified cerebellum DHMRs using $p < 0.05$, as performed above for the liver MeDIP-seq data. We emphasize that relaxing the statistical threshold may lead to some false positive differential enrichments of 5hmC in the cerebellum. Overall, we conclude that differences in 5hmC signal in the liver are more prominent than in the cerebellum. With the relaxed statistical threshold, we obtained 43,153 total DHMRs in the cerebellum, 17,813 with greater enrichment in the old (FC ≥ 2, $p < 0.05$) and 25,340 with greater enrichment in young (FC ≤ −2, $p < 0.05$) (Supplementary Fig. 7B, Supplementary Dataset 10). Verification of the 5hmC signal at the center of the hyper DHMRs showed reproducibly higher signal for old but not young, and vice versa for the hypo DHMRs (Supplementary Fig. 7C). Example genome browser views of hyper and hypo DHMRs in the cerebellum are also shown for individual replicates in Supplementary Fig. 7D. In accord with the idea that age-related differences in 5hmC primarily occur at regions associated with tissue-specific function (Fig. 2E), we observed that the cerebellum DHMRs were enriched for GO terms relating to cerebellum and brain function (for example, dendrite extension and exocytosis of neurotransmitter) (Supplementary Fig. 7E). Annotations of the cerebellum DHMRs were also comparable with the liver DHMRs

and DMRs, wherein most were associated with interCGIs and gene bodies, primarily intronic regions (Supplementary Fig. 7F). In contrast to the liver, we did not observe statistically significant differences in age-related gene body 5hmC signal of all mm10 genes within the cerebellum, despite a trend indicating a potential increase in 5hmC signal with age (Supplementary Fig. 7G).

Consistent with our liver data (Fig. 3A) and previous literature[30−33,35], we observed a positive association between 5hmC and gene expression for both young and old age groups (Supplementary Fig. 8A). We then traced the gene body 5hmC signal over genes with minimal and maximal expression changes with age (Supplementary Dataset 10). We found that genes that remain relatively unchanged in expression with age had significantly higher 5hmC signal in the old, whereas genes that underwent dramatic transcriptional changes with age had no significant differences in 5hmC (Supplementary Fig. 8B). Similar to the liver, genes that underwent maximal transcriptional changes with age tended to harbor significantly shorter 5'UTRs, 3'UTRs, transcript length, CDS length, and had fewer exons compared to the genes that underwent minimal transcriptional changes with age (Supplementary Fig. 8C).

Collectively, these results reiterate that 5hmC undergoes genome-wide changes at gene bodies and regions associated with tissue-specific function and might serve a common transcriptionally restrictive function in mouse tissues during aging.

## Human tissues also show 5hmC-mediated transcriptional restriction
We next investigated whether 5hmC's transcriptionally restrictive function extended to human tissues—thus indicating a potentially conserved epigenetic regulation in aging tissues. Accordingly, we mined publicly available human RNA-seq data from post-mortem human tissues, brain (cortex), heart (left ventricle), and liver, from the Genotype-Tissue Expression (GTEx) project[66] and two published human 5hmC datasets (hmC-CATCH-seq[31] and 5hmC-Seal[30]) for the corresponding tissues (Fig. 7A). Tissue-specific genes were obtained from the Human Protein Atlas, defined as genes with at least 4-fold higher expression in the tissue of interest compared to any other tissues ("tissue enriched") or the average of all other tissues ("tissue enhanced")[67]. We hypothesized that genes that undergo maximal transcriptional changes with age (regardless of direction) would be marked by relatively lower levels of 5hmC, whereas tissue-specific genes (i.e., a combination of tissue enriched and tissue enhanced genes), which are typically enriched for 5hmC, would undergo relatively lower or no detectable transcriptional changes with age.

To mitigate possibilities of disease-related epigenetic and transcriptional changes, we filtered out donors from the GTEx datasets that experienced a slow death, defined by death after a long illness with >1 day of a terminal phase. Sample size distributions by age groups and sex for the GTEx tissues used in this study are shown in Supplementary

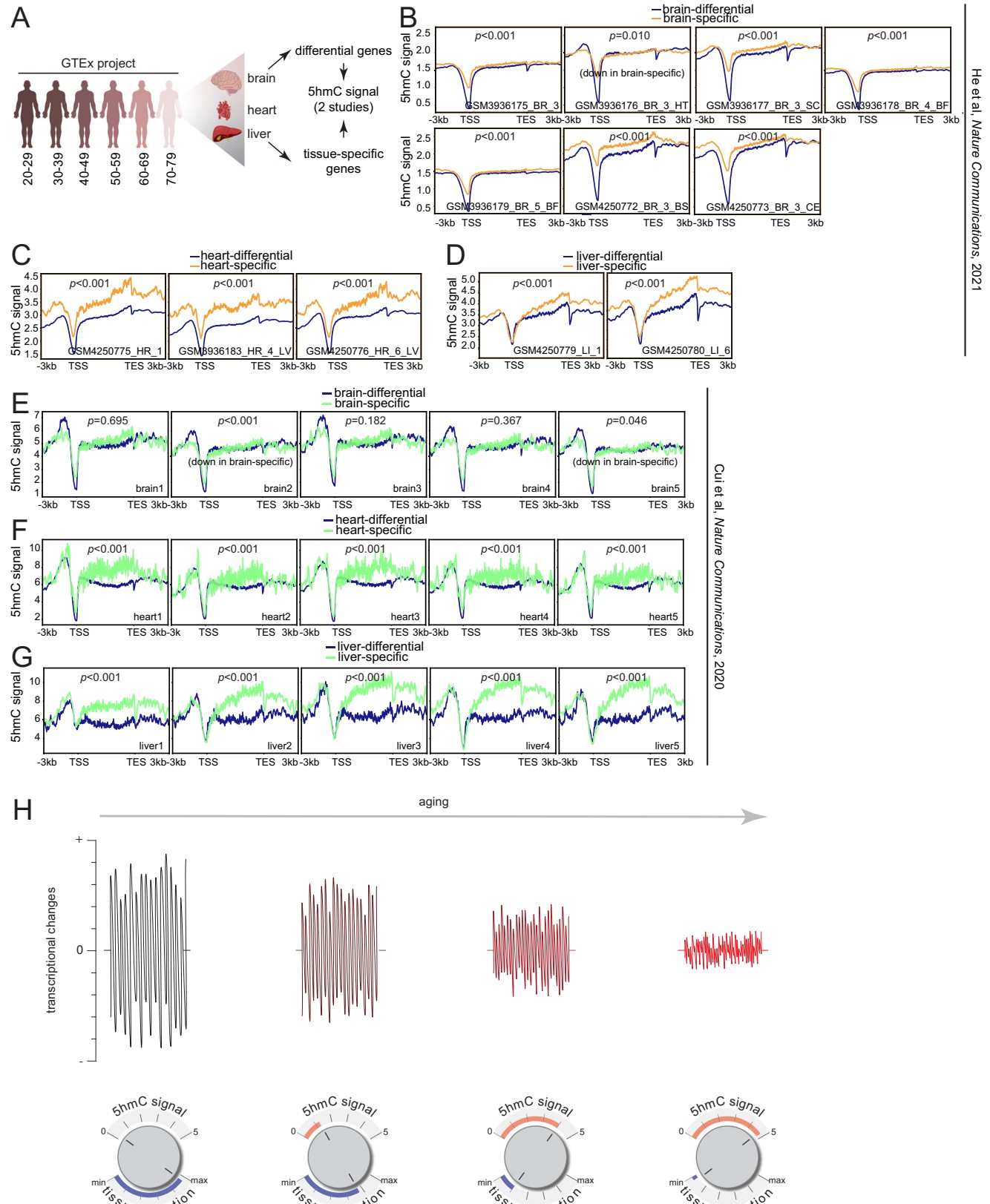

**Fig. 7 | Human tissues also show 5hmC-mediated transcriptional restriction.** **A** Schematic outline of procedure, age groups, and tissues chosen from the GTEx project. **B** Metaplots of 5hmC signal from He et al.[31] over gene bodies of brain-differential and brain-specific genes, **C** heart-differential and heart-specific genes, and (**D**) liver-differential and liver-specific genes; statistical significance was assessed using Mann−Whitney *U* test. **E** Metaplots of 5hmC signal from Cui et al.[30] over gene bodies of brain-differential and brain-specific genes, (**F**) heart-differential and heart-specific genes, and (**G**) liver-differential and liver-specific genes; statistical significance was assessed using Mann−Whitney *U* test. **H** Model illustrating the transcriptionally restrictive role of 5hmC and its propensity to downregulate tissue-specific functions with increasing age. Source data are provided as a Source Data file. Illustration credit: Endosymbiont GmbH.

Fig. 9A. Overall, we included RNA-seq data from 193 brain donors, 393 heart donors, and 207 liver donors. The sex of most donors for each tissue type were male.

To identify genes that undergo significant transcriptional changes during aging, we employed ImpulseDE2[68], an R Bioconductor package designed to model longitudinal datasets, i.e., RNA-seq, and capture permanent and temporal gene expression trajectories. Sex was included as a covariate in all models. For the brain, ImpulseDE2 identified 5,625 genes that underwent significant monotonous expression trajectories with age (5,574 downregulated and 51 upregulated) and one gene that was transiently upregulated with age (Supplementary Fig. 9B, left, Supplementary Dataset 11). Given our interest in the magnitude of transcriptional changes with age, rather than direction, we classified all the age-related permanently and transiently changing genes as "brain-differential" genes. In contrast to the brain-differential genes, "brain-specific" genes did not undergo significant transcriptional changes with age (Supplementary Fig. 9B, middle). As expected, GO analysis of the brain-specific genes revealed biological processes associated with nervous system and synaptic transmission, while the differential genes were enriched for catabolic processes and transport (Supplementary Fig. 9B, right).

In heart tissue, ImpulseDE2 identified 8,163 genes that underwent significant monotonic expression changes with age (8,078 down-regulated, 85 upregulated) and 35 genes with transient expression changes (33 upregulated, 2 downregulated) (Supplementary Fig. 9C, left). We detected no significant transcriptional changes in heart-specific genes with age (Supplementary Fig. 9C, middle). GO analysis showed that the heart-specific genes were enriched for biological processes associated with cardiovascular function, while the heart-differential genes were associated with protein transport, transcriptional and chromatin regulation, and mRNA splicing (Supplementary Fig. 9C, right).

Lastly, in liver tissue, ImpulseDE2 identified 1,852 genes that underwent significant monotonous expression changes with age (1,789 downregulated, 63 upregulated) and 3 genes that underwent transient expression changes (1 upregulated, 2 downregulated). The liver-specific genes also showed no significant changes in expression with age (Supplementary Fig. 9D, middle) and were associated with metabolic processes, while the liver-differential genes were associated with cell division, migration, and shape (Supplementary Fig. 9D, right).

We next sought to assess gene body 5hmC levels of the tissue-specific and tissue-differential genes. Using human 5hmC data obtained from He et al.[31], we observed that genes with significant transcriptional changes in the brain during aging (brain-differential) had relatively lower gene body 5hmC signal compared to the brain-specific genes, with the exception of one sample (Fig. 7B). In heart and liver tissue, we detected significantly higher 5hmC levels in the tissue-specific genes compared to the differential genes (Figs. 7C and 6D). Human 5hmC data from Cui et al.[30] only showed significant differences between brain-specific and brain-differential genes for two samples (Fig. 7E). However, all heart and liver samples showed significantly higher 5hmC levels in the tissue-specific genes compared to the differential genes, in agreement with data from He et al.[31] (Fig. 7F, G). Collectively, these results suggest that 5hmC may be a common regulator of transcriptional restriction in aged mammalian tissues, with significant differences in the heart and liver, and to a lesser extent, in the brain.

### 5hmC is downregulated in response to high-fat diet and disulfiram

Finally, given the potential implications for 5hmC enrichment at regions associated with metabolic processes in the aged liver, we investigated whether levels of this modification were detrimental under stress conditions known to promote metabolic dysfunctions, such as high-fat diet (HFD) consumption. Using C57BL/6 J mice of both sexes, we assessed the effects of HFD (60% kcal fat, $n = 6$) compared to standard diet (SD, 10% kcal fat, $n = 6$) on global 5hmC levels in the liver (schematic, Supplementary Fig. 10A). We observed an overall significant decrease of 5hmC signal in HFD livers (Supplementary Fig. 10B, C); however, the decrease was strikingly sex-specific. Females on HFD had significantly lower 5hmC levels compared to SD, whereas males on HFD had no significant differences in 5hmC levels (Supplementary Fig. 10D). Several studies have reported on the sex-specific differences in metabolic response to HFD[69–71]. In general, females are typically more protected, compared to males, from the immediate adverse effects of HFD. Thus, considering 5hmC's role in restricting the magnitude of transcriptional changes with age, it is tempting to speculate that the downregulation of global 5hmC levels in the females might permit transcriptional flexibility to accommodate the HFD consumption.

We thus inquired whether treatments known to reduce HFD-induced metabolic dysfunction would also alter global 5hmC levels. Disulfiram (DSF), an FDA-approved drug used to treat alcoholism, was previously shown to harbor metabolic protective effects against HFD by reducing weight gain, liver steatosis, and promoting insulin responsiveness[72]. We assessed global 5hmC levels in liver gDNA of four different groups of C57BL/6 J mice of both sexes obtained from Bernier et al.[72]. The mice were fed either a HFD (3 months) then switched to SD (3 months), HFD for the entire duration of the study (6 months), HFD with low dosage (100 mg/kg body weight/day) of DSF (HFD-L, 6 months), or HFD with high dosage (200 mg/kg body weight/day) of DSF (HFD-H, 6 months) (schematic, Supplementary Fig. 10E). Overall, we did not observe significant differences in 5hmC between the SD switch group and the HFD group (Supplementary Fig. 10G), although the latter tended to decrease 5hmC levels. Interestingly, however, HFD-L and HFD-H treatment groups had significantly lower 5hmC signal compared to SD switch and HFD groups. We further assessed whether these effects were sex-specific and observed a stronger dose-response relationship for the DSF as well as significantly lower 5hmC in the DSF groups compared to SD and HFD groups in the males (Supplementary Fig. 10G, right). Females, however, had comparable levels of 5hmC among the HFD, HFD-L, and HFD-H treatment groups. Altogether, these results suggest that altered 5hmC levels may be a potential mechanism underlying sex-specific response to HFD and that downregulation of the modification may have long-term beneficial effects against HFD, possibly by promoting transcriptional flexibility.

## Discussion

We have surveyed global 5mC/5hmC levels and found that 5hmC accumulates in the aged liver without detectable global differences in 5mC (Fig. 1). We performed genome-wide profiling of 5hmC in mouse liver and cerebellum and identified specific increases in 5hmC at gene bodies associated with tissue-specific function (Fig. 2 and Supplementary Fig. 7), in agreement with previous studies showing that 5hmC marks tissue-specific genes[30,31,35]. Surprisingly, we observed that the age-related accumulation of 5hmC is not driven directly by differences in expression or enzymatic activity of TET proteins, but rather age-related contexts such as prolonged quiescence and senescence (Fig. 5 and Supplementary Fig. 5). We further observed that regions gaining 5hmC with age were marked by relatively higher 5mC levels in the young (Supplementary Fig. 3C, right), consistent with a progressive conversion of 5mC to 5hmC over time.

It is unclear why age-related differences in 5hmC occur primarily at tissue-specific genes. A proposed mechanism for 5hmC's tissue-specific localization posits that tissue-specific TFs recruit TET proteins to tissue-specific genes, which then traverse the gene body alongside transcription elongation complexes to oxidize 5mC to 5hmC[73,74]. Indeed, we identified TFs from Lisa analysis (Supplementary Fig. 1H) that bind genes with high 5hmC signal and are known TET interactors. For example, PPARγ has been previously shown to interact with TET1

and direct the local increase of 5hmC levels[75]. Direct evidence for TET interactions with the other TFs is currently lacking, however, knockdown of ZIC2 has been shown to decrease the binding affinity of TET1 in primed epiblast[76]. Knockdown of Glis2 has also been shown to decrease 5hmC levels in human embryonic stem cells[77], while Tbx3 over-expression results in higher 5hmC levels[78]. These data suggest that 5hmC's enrichment in tissue-specific genes in the liver may be due to recruitment of TET proteins by TFs. Our data thus provides preliminary evidence for a proposed model wherein TET proteins interact with tissue-specific TFs, resulting in 5hmC localization to tissue-specific genes[73,74] (Supplementary Fig. 1H).

Given that a functional role for 5hmC in aging is largely unknown, our primary focus in this study was to dissect the role of 5hmC in transcriptional changes during aging. We found that although 5hmC is typically enriched in gene bodies of transcriptionally active and tissue-specific genes, the modification has a conserved function to restrict the magnitude of transcriptional changes during aging (Figs. 3, 6, 7, Supplementary Figs. 8 and 9).

5hmC is catalyzed from 5mC and we indeed observed opposing 5mC/5hmC levels at genes with minimal gene expression changes during aging (Fig. 3F, G), however the changes in 5mC were modest and not statistically significant, suggesting that changes in 5hmC may be a better predictor of age-related transcriptional changes. In support of our finding, Tet1/3 double-deficient mice have been shown to exhibit lower transcriptional fidelity in early embryogenesis[79]. Suggestively, transcriptional dysregulation is a hallmark of cancer, an age-related disease with extensively documented TET mutations as well as loss of 5hmC[80–82]. Perhaps the function of 5hmC may be primarily protective, maintaining stable gene expression with age. However, we also speculate that 5hmC may need to be downregulated to enable proper transcriptional responses under certain conditions such as HFD and drug treatment (Supplementary Fig. 10). Therefore, while the mark promotes transcriptional restriction during aging to preserve tissue function, it may also hinder the transcriptional flexibility required during stress.

We recall above that another epigenetic modification, H3K36me3, has been previously shown to restrict the magnitude of age-related transcriptional changes in *C. elegans* and *D. melanogaster*, two model organisms that generally lack (or have undetectable levels of) 5hmC[48]. Both H3K36me3 and 5hmC are typically localized at gene bodies, though H3K36me3 levels have been reported to decline with age[83]. Possibly 5hmC may have evolved as an additional regulator of transcriptional restriction in more complex organisms.

We investigated 5hmC-interacting proteins and identified several factors involved in translation, transcription, and chromatin accessibility (Fig. 4B, C, Supplementary Figs. 4B and 4C). Interestingly, we observed that 5hmC has lower binding affinity for splicing-associated factors (Fig. 4D, E) and is positively associated with age-related splicing events (Fig. 4F, G). Similarly, genes with minimal expression changes with age exhibited higher age-related splicing events, as shown by both short-read (Fig. 4F) and long-read (Fig. 4G) sequencing. We speculate that 5hmC might exert its transcriptionally restrictive function during aging through splicing-associated factors; future work aims to test this notion.

Lastly, we showed that increases in 5hmC are also associated with lower gene expression in old samples (Fig. 3C, D, Supplementary Fig. 1I). We demonstrate this by leveraging 5hmC's tissue-specific localization and link 5hmC production to downregulation of ATP production in HepG2 cells (Fig. 5C–E, G–I).

Overall, our work elucidates an important and previously unrecognized function of a relatively understudied epigenetic modification, 5hmC, in aging. Aging is generally associated with global decline of cellular and organ function and accompanied transcriptional changes. While 5hmC aids in maintaining homeostasis within an aged environment by imposing transcriptional restrictions on tissue-specific genes,

prolonged periods of elevated 5hmC levels may potentially down-regulate tissue-specific functions (Fig. 7H). Additionally, heightened levels of 5hmC might prove detrimental in the face of stress and requires a reduction to enable appropriate transcriptional responses in reaction to environmental cues.

A limitation in our study is the incapacity to explore sex differences during aging due to a small sample size ($n = 2$ per sex in each age group), despite observing some sex-specific variations in 5hmC (Fig. 2A) and mRNA levels (Supplementary Fig. 4H). These preliminary observations merit further investigation in future studies.

## Methods

### Animals

This study was approved by the Animal Care and Use Committee of the NIA in Baltimore, MD under Animal Study Protocol number 481-LGG-2022 (all except DSF experiments) and 444-TGB-2016 (DSF experiments). Young and old inbred C57BL6/JN mice of both sexes were acquired from the NIA aged rodent colony (https://ros.nia.nih.gov/) and housed in rooms that were maintained at $22.2 \pm 1\,°C$ and 30–70% humidity. The HFD experiments were performed in Jackson labs. For DSF experiments, C57BL/6 J mice (Jackson Laboratory stock #000664) were single housed under temperature-controlled conditions with 12 h light/12 h dark cycle with ad libitum access to house chow (2018 Teklad Global 18% Protein Rodent Diet 2018S, Harlan Teklad) and water. Routine tests were performed to ensure that mice are pathogen-free and sentinel cages maintained and tested according to American Association for Accreditation of Laboratory Animal Care (AAALAC) criteria. The age and sex information are available in Supplementary Dataset 1.

### Cell lines and culture conditions

HepG2 (ATCC, human male) cells were cultured in a 37 °C 5% $CO_2$ and 20% $O_2$ humidified incubator with Dulbecco's Modified Eagles Medium (DMEM, Gibco) supplemented with 10% Fetal Bovine Serum (FBS, Thermo Fisher) and 1% penicillin/streptomycin (Penn/Strep, Thermo Fisher). WI-38 (Coriell Institute, human female) cells were cultured in a 37 °C 5% $CO_2$ and 20% $O_2$ humidified incubator with DMEM (Gibco) supplemented with 10% heat-inactivated FBS (Gibco), 0.5% Penn/Strep (Gibco), sodium pyruvate (Gibco), and non-essential amino acids (Gibco). We also mined hMeDIP-seq and RNA-seq data from Peng et al.[64] who used T24 (human male) bladder carcinoma cells with and without vitamin C treatment.

### High-fat diet and disulfiram treatment

For mice on HFD, two regimens were used. In the first regimen, the diet was applied early in life and for a short duration. C57BL/6 J male and female mice ($n = 3$ each, Jackson Laboratory stock #000664) at 6 weeks of age were fed with Research Diets, Inc. D12492i (60 kcal% fat) diet up to 11 weeks of age. An equal number of control mice of the same genotype and sex were fed with Research Diets, Inc. D12450Bi (10 kcal% fat) up to 11 weeks of age. Mice were then sacrificed, and their livers harvested. In the second regimen, diet was applied later and for longer duration. Beginning at 9 months of age, mice were maintained on a HFD consisting of AIN-93G modified to provide 60% of calories from fat (HFD; carbohydrate:protein:fat ratio of 16:23:61) for the next 3 months, after which animals were randomly divided into four groups ($n = 9$, 6 males and 3 females). Group 1, was continued on HFD diet; Group 2, was switched to standard AIN-93G diet (SD, carbohydrate:protein:fat ratio of 64:19:17), Group 3 was fed HFD supplemented with a low dose of DSF (100 mg/kg body weight/day DSF; HFDL); and Group 4 was fed HFD supplemented with a high dose of DSF (200 mg/kg body weight/day DSF; HFDH). All animals were sacrificed after an additional 3 months of treatment and livers were collected for further analysis. DSF (Sigma-Aldrich, St-Louis, MO) was included in HFD-modified AIN-93G diet (Dyets, Inc., Bethlehem, PA) at a concentration

of 2.33 g/kg (low dose, HFDL) and 2.67 g/kg (high dose, HFDH), respectively.

### Partial hepatectomy surgery

70% partial hepatectomy was performed in accordance with the guidelines from Mitchell et al.[84] and previously reported in Yang et al.[47]. Briefly, liver lobes were removed and labeled as "pre-surgery". After indicated post-surgery time, animals were sacrificed by carbon dioxide asphyxiation and cervical dislocation. The liver was dissected and frozen in isopentane chilled with liquid nitrogen and stored in −80 °C.

### Induction of quiescence

To induce quiescence of HepG2 cells by serum starvation, ~70% confluent plates of HepG2 cells were changed to DMEM without FBS for 1–3 days. To induce quiescence by contact inhibition, cells were allowed to grow until they reached 100% confluency. The cultures were maintained for 2, 7 or 14 days with media change (with serum) every two days. Proliferating controls were included for both serum starvation and contact inhibition experiments.

### Induction of senescence

WI-38 cells were maintained at low population doubling (PD) levels for proliferating conditions. Cellular senescence was triggered by different methods. ETIS was achieved by culturing for 10 days in the presence of etoposide (Selleckchem) at 50 μM, with medium refreshed every 3 days. IRIS was achieved by exposing cells to 15 Gray (Gy) followed by culturing for 10 days. OSIS was achieved by adding 0.75 mM $H_2O_2$ directly to cells in complete medium and replacing with fresh medium 2 h later.

### RT-qPCR for senescent cells

Cells were lysed in either Tri-Reagent (Invitrogen) or RLT buffer (Qiagen), and the lysate was processed with the QIAcube (Qiagen) to purify total RNA, which was then reverse-transcribed (RT) to create cDNA using Maxima reverse transcriptase (Thermo Fisher Scientific) and random hexamers. Real-time, quantitative (q)PCR analysis was then performed using SYBR Green mix (Kapa Biosystems), and the relative expression was determined by the 2-ΔΔCt method on a QuantStudio 6 qPCR machine (Thermo Fisher). The levels of mRNAs were normalized to human *Actb*.

### BrdU assay

~125,000 cells were seeded in 6-well plates and incubated with BrdU diluted in DMEM with 10% FBS for 24 h. BrdU incorporation was measured following the manufacturer's protocol (Cell Signaling Technology). Briefly, cells were fixed and denatured before the addition of anti-BrdU mouse monoclonal antibody. BrdU incorporation was detected by measuring absorbance at 450 nm using a GloMax plate reader (Promega).

### Induction and neutralization of ROS in cell culture

To induce acute ROS, ~1 × 10⁶ HepG2 cells were treated with 600 μM $H_2O_2$ (Sigma) in serum-free DMEM for 2 h then changed into fresh media (without serum) for 24 h. To neutralize ROS, HepG2 cells were either co-treated or sequentially treated with $H_2O_2$ and N-acetylcysteine amide (NAC). For co-treatment, cells were first treated with 600 μM $H_2O_2$ and 2 mM NAC in serum-free DMEM for 2 h, then washed twice and incubated with 2 mM NAC in serum-free DMEM for 24 h. Sequential treatment was performed by first treating cells with 600 μM $H_2O_2$ in serum-free DMEM for 2 h, then washed twice and incubated with 2 mM NAC in serum-free DMEM for 24 h. DHE staining was also performed in parallel for the ROS neutralization experiments, using separate wells, as described in the "ROS detection in cells and tissue sections" below. To induce chronic ROS, ~1 × 10⁶ HepG2 cells were treated with 20 μM $H_2O_2$ in serum-free DMEM for 24 h. As control,

cells were kept in serum-free DMEM with DMSO for the duration of treatment.

### Treatment with Vitamin C

To induce production of 5hmC by vitamin C, ~1 × 10⁶ HepG2 cells were treated with vitamin C (Sodium L-ascorbate, Sigma) at concentrations of 30, 75, 150, and 300 μM in DMEM with serum for 24 h.

### Genomic DNA isolation for dot blots

gDNA was isolated from ~25 mg of frozen tissue or ~1 × 10⁶ HepG2 cells suspended in 200 μL of PBS using the Quick-DNA Miniprep Plus kit (Zymo Research) following the manufacturer's protocol with an overnight (for frozen tissue) or 10 min (for cells) proteinase K digestion. The amount and quality of extracted DNA was assessed using Qubit HS assay kit (Thermo Fisher) and NanoDrop One (Thermo Fisher).

### Quantification of cytosine modification by mass spec

gDNA was isolated from ~25 mg tissue following instructions on the Quick-DNA Miniprep Plus Kit using an overnight proteinase K digestion. The amount of DNA was quantified using a Nanodrop and samples were verified to have a 260/280 of >1.8 and a 260/230 of ≥2. ~2.5 ug of DNA in a volume of 130 μL was sheared to ~800 bp using a S220 focused ultrasonicator (Covaris) and the following parameters: peak incident power 105, duty factor 5%, cycles per burst 200, treatment time: 50 s. The shearing was verified by phenol-chloroform purification followed by ethanol precipitation and running on a 1% agarose gel.

Cytosine methylation and hydroxymethylation were quantified using a protocol modified from Sun et al.[85,86]. Briefly, DNA from samples indicated in (Supplementary Dataset 1) was digested into single nucleosides by using the Nucleoside Digestion Mix (New England Biolabs) enzyme cocktail at 37 °C for 2 h. 5mC was identified and quantified by nLC-MS/MS. Using a Dionex RSLC Ultimate 3000 (Thermo Scientific, San Jose, CA, USA), nLC was configured with a 300 μm ID x 0.5 cm C18 trap column (Dionex, Thermo Scientific) and a 75 μm ID x 25 cm Reprosil-Pur C18-AQ (3 μm; Dr. Maisch GmbH, Germany) analytical nano-column were used to identify and quantify the absolute value of 5mC and 5hmC as shown in Fig. 1A. nLC was configured with a two-column system consisting of a 75 μm ID x 1 cm polygraphitic carbon resin (PGC, HyperCarb, Thermo Scientific) trap column and a 75 μm ID x 25 cm PGC analytical nano-column for relative 5hmC comparison between young and old ($n = 3$ each) mice in liver samples shown in Fig. 6B. Except for the C18 trap column (cartridge from Thermo Scientific), all other columns were packed in-house. nLC was coupled online to an Orbitrap Fusion Lumos mass spectrometer (Thermo Scientific). The spray voltage was set to 2.3 kV and the temperature of the heated capillary was set to 275 °C. PGC columns setup was used for relative 5mC and 5hmC analysis. The full scan range of 110 – 1200 m/z was acquired in the Orbitrap at a resolution 120,000. Targeted scans were performed for MS/MS fragmentation using an HCD energy of 30 V and acquired in the Orbitrap at a resolution of 7,500. To accurately quantitate the absolute value, C18 columns are used, and the instrument was optimized to fragment the protonated nucleosides deoxycytidine (dC), deoxy-methylcytidine (dmC) and deoxy- hydroxymethylcytosine (dhmC) into protonated nucleobases C, 5mC, and 5hmC with m/z at 112.0505, 126.0662, and 142.0611, respectively. The full scan range was 110 – 600 m/z acquired in the Orbitrap at a resolution 120,000. The source fragmentation energy was set at 30 V and RF lens % was set at 50, which gives >90% generation of nucleobases. To accurately quantify 5mC and 5hmC, a calibration curve of 5mC% and 5hmC% were constructed by analyzing samples with varying amount of 5mC and 5hmC standard in the presence of constant C standard and were used to correct the observed 5mC% and 5hmC% from the real samples. Quantification was obtained by extracting the ion chromatograms of C, 5mC and 5hmC using

Skyline software. 5mC and 5hmC levels were calculated by dividing the area under the curve of the given species by the total area of all (un)modified C quantified.

## Immunofluorescence

Fresh liver tissues were fixed with 4% methanol-free formaldehyde at 4 °C overnight and then immersed in 20% sucrose solution at 4 °C overnight. The tissues were embedded in OCT compound, frozen at −80 °C, and then cut into 12 μm sections onto positively charged slides in a cryostat chamber. The sections were permeabilized with 0.2% Triton X-100 in Tris Buffered Saline (TBS) for 5 min at room temperature. Antigen retrieval steps were performed based on previous publication[87]. Briefly, sections were treated with 2 N hydrochloric (HCl) acid in PBS for 30 min in a 37 °C incubator. After denaturation, sections were neutralized in two successive rounds with 0.1 M Tris-HCl (pH: 7.5) in PBS for 5 min. The sections were blocked for 1 h at room temperature with 2% normal goat serum (Vector Biolabs) then incubated with 2 μg/mL dilution of 5hmC antibody (Active Motif, 39092) overnight at 4 °C in a humidified chamber. After three rounds of washes with TBS (supplemented with 0.1% Tween-20, TBST; Pierce), the sections were incubated with a secondary antibody conjugated to a fluorescent dye (Thermo Fisher, A-11008) for 1 h at room temperature. Lastly, sections were stained with 5 μg/mL DAPI in TBS for 1 h at room temperature in a humidified chamber. Following washes with TBS, the sections were mounted with Epredia Lab Vision PermaFluor Aqueous Mounting Medium (Fisher Scientific) then photographed using a Zeiss LSM 710 confocal microscope. Intensities were quantified using ImageJ v1.51 g[88].

## Dot blot

The dot blot assay was adapted from a previous publication[15]. 2 μL of gDNA (containing indicated amounts of DNA) was denatured with 0.5 N NaOH for 15 min then spotted on a nitrocellulose blotting membrane (GE Healthcare) and cross-linked at 120,000 microjoules for 20 min using a Stratalinker® UV Crosslinker 1800 (Stratagene). As a negative control, we included 2 μL of water. Unless otherwise specified, 2 μL containing indicated amounts of gDNA isolated from a young mouse hippocampus was used as positive control. The membrane was then blocked in 5% skimmed milk in TBS containing 0.1% Tween 20 (TBST) for 1 h at room temperature followed by incubation with 1:10,000 dilution of 5hmC antibody (Active Motif, 39069) overnight at 4 °C. After three rounds of washes, the membrane was incubated with 1:10,000 dilution of HRP-conjugated anti-rabbit IgG (BioRad, 1706515) for 1 h at room temperature. Digital ECL substrate solution (Kindle Biosciences) was added to the membrane before detection with a ChemiDoc MP Imaging System (BioRad). The dot blot intensity was quantified using ImageJ v1.51 g[88].

## ROS detection in cells and tissue sections

Fresh liver tissues were embedded in OCT compound without fixation, frozen at −80 °C, then cut into 12 μm sections onto positively charged microscope slides (Fisher Scientific) in a cryostat chamber. Slides were rinsed with $H_2O$ for 30 s then incubated with 50 μM dihydroethidium staining solution (DHE, Thermo Fisher) in 1x PBS at 37 °C for 30 min in a dark humidified chamber. Slides were washed twice then stained with 5 μg/mL DAPI in 1x PBS for 30 min at room temperature. The sections were mounted with Epredia Lab Vision PermaFluor Aqueous Mounting Medium (Fisher Scientific) then photographed using a Zeiss LSM 710 confocal microscope. Intensities were quantified using ImageJ v1.51 g[88].

For ROS detection in cells, 70–80% confluent HepG2 cells in μ-Slides (Ibidi, 80826) were treated with $H_2O_2$ (Sigma) in serum-free DMEM for 2 h at concentrations of 0, 10, 50, 100, 200, and 600 μM. Cells were washed twice with PBS and then incubated with 8 μM of DHE (Thermo Fisher) in serum-free DMEM for 15 min at 37 °C. After washes, cells were incubated with DAPI (1:1000) at 37 °C for 30 min and then photographed using a Zeiss LSM 710 confocal microscope.

## ATP assay

Proliferating, quiescent (contact-inhibited for 14 days), or $H_2O_2$ treated HepG2 cells ($n = 3$ biological replicates) were trypsinized, counted, and $1 × 10^6$ cells were pelleted. The cell pellets were resuspended in 350 μL DMEM medium, and 100 μL was distributed to 3 wells of a 96-well plate for each sample (technical replicates, $n = 3$ per biological replicates). 100 μL of CellTiter-Glo reagent (Promega) was then added to each well containing cell suspension. The contents were mixed for 2 min on an orbital shaker to induce cell lysis. The plate was then incubated at room temperature for 10 min to stabilize the luminescent signal and recordings were taken on a GloMax Discover System (Promega).

## TMRM assay

After contact inhibition or treatment of HepG2 cells with vitamin C, cells were resuspended in 1 mL PBS at -$1 × 10^6$ cells/mL. TMRM was added to a final concentration of 20 nM and incubated for 30 min at 37 °C, 5% $CO_2$. For CCCP control samples, CCCP was added to a final concentration of 50 nM to the cells, incubated for 5 min at 37 °C, 5% $CO_2$ and then treated with 20 nM TMRM reagent for 30 min. Cells were analyzed on a BD Symphony flow cytometer with 561 nm excitation.

## Nuclei preparation

Nuclei preparations were performed by douncing frozen liver tissue in nuclei preparation buffer containing 10 mM Tris-HCl (pH 7.4), 10 mM NaCl, 3 mM $MgCl_2$, 0.1% Tween 20, 0.1% NP-40, 0.01% digitonin, 1 mM BSA, and supplemented with 1x Halt protease and phosphatase inhibitor cocktail (Thermo Fisher) and 1 mM sodium butyrate. The resulting homogenate was filtered through a 30 μm cell strainer then washed using wash buffer containing 10 mM Tris-HCl (pH 7.4), 10 mM NaCl, 3 mM $MgCl_2$, 0.1% Tween 20, 1% BSA, and supplemented with 1x Halt protease and phosphatase inhibitor cocktail (Thermo Fisher) and 1 mM sodium butyrate to stop lysis. After centrifugation, the nuclei pellet was washed thrice in nuclei suspension buffer containing PBS, 2% BSA, 3 mM $MgCl_2$, and supplemented with 1x Halt protease and phosphatase inhibitor cocktail (Thermo Fisher) and 1 mM sodium butyrate.

## TET activity assays

Nuclei preparations were made as mentioned above. The nuclei pellets were lysed in nuclei lysis buffer containing 10 mM Tris-HCl (pH 7.4), 100 mM NaCl, 1 mM EDTA, 0.5 mM EGTA, 0.1% sodium-deoxycholate, 0.5% N-lauroylsarcosine, and supplemented with 1x Halt protease and phosphatase inhibitor cocktail (Thermo Fisher) and 1 mM sodium butyrate, and then sheared to <500 bp using a Covaris S220 Ultrasonicator (peak incident power 140, 200 cycles per burst, duty factor 5%, 10 min). The protein was quantified using the Pierce™ BCA Protein Assay Kit (Thermo Fisher) and ~10 μg total protein was used to measure TET activity using the TET Hydroxylase Activity Quantification Kit (Abcam) following the manufacturer's protocol.

## DNA oligo pulldown mass spectrometry

Nuclei preparations were made as mentioned above. Following a published protocol[29], the nuclei pellet was then lysed on ice for 90 min in 2 volumes of nuclei lysis buffer containing 420 mM NaCl, 20 mM HEPES, 20% v/v glycerol, 2 mM $MgCl_2$, 0.2 mM EDTA, 0.1% NP40, 0.5 mM DTT, and supplemented with 1x Halt protease and phosphatase inhibitor cocktail (Thermo Fisher) and 1 mM sodium butyrate. Prior to protein quantification, samples were pre-cleared using washed Dynabeads MyOne C-1 beads (Thermo Fisher) suspended in nuclei lysis buffer. DNA pull-downs were performed as described in ref. 29 with 10 ug of DNA oligo for each pull-down (unmodified, methylated, and hydroxymethylated; GenScript) and 400 ug of nuclear extract. 10% lysate was saved as "input".

Proteins were eluted from beads at room temperature with mixing for 20 min and at 65 °C for 10 min on a thermomixer (800 RPM) in

two successive rounds using 100 μL of biotin elution buffer (12.5 mM D-biotin, 7.5 mM HEPES pH 7.5, 75 mM NaCl, 1.5 mM EDTA, 0.15% SDS, 0.075% sarkosyl, and 0.02% sodium deoxycholate). The two eluents were pooled and precipitated overnight at 4 °C with chilled trichloroacetic acid (25% v/v). Proteins were pelleted at 16,000 g in 4 °C for 30 min. The supernatant was removed, the pellet washed once with ice-cold acetone, then centrifuged again at 16,000 g in 4 °C for 5 min and air dried for 1 min. An S-Trap column cleanup was performed by loading the protein pellet resuspended in 165 μL of added loading buffer (90% methanol and 10 mM sodium bicarbonate pH 8.0) onto an S-Trap Micro Spin Column (Protifi). The column was washed twice with 150 μL of loading buffer and gently centrifuged before overnight digestion at 37 °C with 0.1 μg/μL of trypsin (Promega) in 50 mM ammonium bicarbonate. After a gentle centrifuge, peptides were first eluted with 40 μL of 0.2% aqueous formic acid then eluted again with 35 μL of 50% acetonitrile containing 0.2% formic acid. Eluted peptides were dried in DNA120 SpeedVac (Thermo Fisher) with no heat.

Prior to mass spectrometry analysis, samples were desalted using a 96-well plate filter (Orochem) packed with 1 mg of Oasis HLB C-18 resin (Waters). Briefly, the samples were resuspended in 100 μL of 0.1% trifluoroacetic acid (TFA) and loaded onto the HLB resin, which was previously equilibrated using 100 μL of the same buffer. After washing with 100 μL of 0.1% TFA, the samples were eluted with a buffer containing 70 μL of 60% acetonitrile and 0.1% TFA and then dried in a vacuum centrifuge.

Samples were resuspended in 10 μL of 0.1% TFA and loaded onto a Dionex RSLC Ultimate 300 (Thermo Scientific), coupled online with an Orbitrap Fusion Lumos (Thermo Scientific). Chromatographic separation was performed with a two-column system, consisting of a C-18 trap cartridge (300 μm ID, 5 mm length) and a picofrit analytical column (75 μm ID, 25 cm length) packed in-house with reversed-phase Repro-Sil Pur C18-AQ 3 μm resin. To analyze the proteome, peptides were separated using a 60 min gradient from 4–30% buffer B (buffer A: 0.1% formic acid, buffer B: 80% acetonitrile +0.1% formic acid) at a flow rate of 300 nL/min. The mass spectrometer was set to acquire spectra in a data-dependent acquisition (DDA) mode. Briefly, the full MS scan was set to 300–1200 m/z in the orbitrap with a resolution of 120,000 (at 200 m/z) and an AGC target of 5x10e5. MS/MS was performed in the ion trap using the top speed mode (2 s), an AGC target of 1x10e4 and an HCD collision energy of 35.

Proteome raw files were searched using Proteome Discoverer software (v2.4, Thermo Scientific) using SEQUEST search engine and the SwissProt mouse database. The search for total proteome included variable modification of N-terminal acetylation, and fixed modification of carbamidomethyl cysteine. Trypsin was specified as the digestive enzyme with up to 2 missed cleavages allowed. Mass tolerance was set to 10 ppm for precursor ions and 0.2 Da for product ions. Peptide and protein false discovery rate was set to 1%. Following the search, data was processed as described previously[89]. Briefly, protein abundances were $\log_2$ transformed and normalized by the average value of each sample. Missing values were imputed using a normal distribution set 2 standard deviations below the mean. Statistical differences were assessed using Welch's $t$-test (if the F-test $p$-value was <0.05), otherwise the standard Student's $t$-test was used.

## RNA isolation, RT-qPCR, and RNA-sequencing
Liver RNA-seq data was previously generated and reported in Yang et al.[47]. For cerebellum, RNA was isolated from frozen tissue by homogenization in Trizol followed by isopropanol precipitation. The RNA was further purified using RNeasy columns (Qiagen). An on-column DNase I digestion was performed during the purification step to remove gDNA. The RNA amount and integrity were confirmed using the Qubit RNA HS Assay Kit and RNA IQ Assay (Thermo Fisher) respectively. Total RNA (~700 ng) was used to make RNA-seq libraries following the Zymo-Seq Ribo-free Total RNA Library Kit (Zymo

Research) instructions with dual indexing. The RNA-seq libraries were pooled into equimolar amounts, further quantified using the NEBNext Library Quant Kit (New England Biolabs), and then subjected to two rounds of 50 bp paired end sequencing on a NextSeq 2000 platform using a P2 100-cycle kit (Illumina).

## Direct RNA-seq with nanopore sequencing
RNA integrity was assessed using Qubit RNA IQ assay kit (Thermo Fisher) before library preparation. 25–50 μg of total RNA was used to make libraries using the direct RNA sequencing kit (Oxford Nanopore Technologies) as previously described with modifications[90]. Briefly, after selection of poly(A) RNAs using Oligo d(T)25 Magnetic Beads (New England Biolabs), 15 pmoles of REL5 adapter (/5Bio/rArA rUrGrArUrArCrGrGrCrGrArCrCrArCrCrGrArGrArUrCrUrArCrArCrUrC rUrUrUrCrCrUrArCrArCrGrArCrGrCrUrCrUrUrCrCrGrArUrCrU) was ligated to the 5′ ends of poly(A)-purified RNAs using T4 RNA ligase 1 (New England Biolabs) for 3 h at 37 °C. 750 ng of REL5-ligated poly(A) RNAs was used for library preparation according to manufacturer's protocol (Oxford Nanopore Technologies). Final libraries were quantified using Qubit 1X dsDNA High Sensitivity (HS) assay kit (Thermo Fisher) and sequenced on a MinION device using R9.4.1 flow cells (Oxford Nanopore Technologies).

## Methyl and hydroxymethyl DNA immunoprecipitation sequencing (MeDIP-seq and hMeDIP-seq)
The liver MeDIP-seq data was previously generated and reported in Yang et al.[47]. The hMeDIP assay was performed using the MagMeDIP-seq Package (Diagenode) following the manufacturer's protocol with a mouse antibody against 5hmC (Diagenode, C15200200-50); 1.2 μg of gDNA was sonicated into ~ 200 bp fragments using the S220 focused ultrasonicator (Covaris). To maximize IP yield for the liver hMeDIP-seq, samples were processed in duplicates then pooled before IPure purification. Prior to immunoprecipitation, samples were spiked with hydroxymethylated and unmethylated internal DNA controls. IP efficiency and success was verified by qPCR targeting internal DNA controls. The DNA amount was quantified by Qubit HS DNA kit (Thermo Fisher) and the fragment size was assessed on a 2100 BioAnalyzer using a DNA HS kit (Agilent). Individual libraries for immunoprecipitated DNA and 10% input were dual indexed (NEBNext Multiplex Oligos, unique dual indices, New England Biolabs), PCR amplified, and then pooled into equimolar amounts and further quantified using the NEBNext Library Quant Kit (New England Biolabs). The pooled library was subjected to 50 bp paired-end sequencing on the Illumina NextSeq 2000 platform using a P2 100-cycle kit (Illumina).

## Antibodies and oligos
All antibodies and oligos used in this study are listed in Supplementary Dataset 12.

## Bioinformatic analysis
**hMeDIP-seq and MeDIP-seq analysis.** hMeDIP-seq and MeDIP-seq sequencing reads were de-multiplexed using bcl2fastq/2.20.0 and adapter trimmed using trimgalore/0.6.6. FastQ quality was assessed using FastQC/ 0.11.9[91]. Reads were aligned to the mouse reference genome (assembly GRCm38/mm10) using bowtie/2-2.4.4 then filtered for a minimum mapping quality of 10 using samtools/1.9[92]. BAM files were then sorted and additionally filtered for uniquely mapped and non-duplicate reads using sambamba/0.7.1[93]. Encyclopedia of DNA Elements (ENCODE) blacklisted regions (mm10)[94] were filtered from BAM files using bedtools/2.30.0. RPKM (reads per kilobase per million mapped reads) normalized bigWig files were generated by first indexing BAM files using samtools/1.9 followed by conversion with the bamCoverage function of deeptools/3.5.0[95]. 5hmC samples were input subtracted using the bigWigCompare function of deeptools/3.5.0. To identify differentially hydroxymethylated/methylated regions, we

employed the R Bioconductor package QSEA/1.26.0[40], using a window size of 400 bp. Within QSEA, we accounted for CpG density per fragment for each 400 bp window using the "blind calibration" method. Random data was generated (Supplementary Fig. 1I, panel 4) by using the R seq function with ranges comparable to young and old gene body 5hmC signal (−24.74 to 58.90, increments of .01) and then randomly sampled with replacement using the R sample function.

**RNA-seq analysis.** Sequencing reads were processed as reported elsewhere[47]. Briefly, reads were de-multiplexed using bcl2fastq/2.20.0, adapter trimmed using trimmomatic/0.39[96], and then quality assessed using FastQC/0.11.9[97]. Reads were aligned to the mouse reference genome, GRCm38/mm10, using STAR/2.7.5b[98] and the resulting BAM files were sorted and indexed using samtools/1.10 [92]. BAM files were filtered for duplicates and alignments with a minimum mapping quality of 10 using picard/2.20.8 and samtools/1.10 [92], respectively. Gene counts were estimated using the featureCounts function of the Rsubread R package/2.6.4[99]. DESeq2/1.30.1[100] was used to perform count normalization and differential gene expression analysis between old and young samples. Genes were ranked and categorized according to DESeq2 normalized mRNA levels (averaged within groups) or absolute FC ratio |old vs young| using the ntile function of dplyr/1.0.7 (Fig. 3B).

To identify age-related alternative splicing events, reads were aligned to the mouse reference genome (GRCm38/mm10) using STAR/2.7.5b[98] with additional parameters (--outSAMstrandField intronMotif --outSAMattrIHstart 0 −alignSoftClipAtReferenceEnds No). rmats/4.1.1[56] was employed on BAM files with parameters --readLength 37 --variable-read-length and --cstat 0.0001.

**Nanopore sequencing data analysis.** Nanopore dRNA-seq data were basecalled using Guppy/6.1.2. Reads were subsequently mapped to mouse genome GRCm38/mm10 using minimap2/2.24[101] with parameters -a -x splice -k 12 -u b −secondary = no. Basecalled reads were also separately aligned against the mouse transcriptome (Ensembl version 92) using -a -x map-ont -k 12 -u f −secondary = no. FLAIR/v1.7.0[102] was used to identify and quantify novel transcripts. To create a unified database of existing and newly identified transcripts all sequenced samples were pooled at the flair collapse step, as suggested by the authors. Internally, DESeq2[103] and DRIMSeq[104] were used for differential expression and differential isoform usage calculations respectively. Poly(A) tail lengths were extracted from sequenced reads using the nanopolish poly(A) package[105]. Only poly(A) tail lengths that passed the software quality control scores and that were tagged as "PASS" were used in our analysis.

To identify transcripts with systematic whole-molecule or poly(A) length changes across experimental conditions, we employed in-house scripts that use linear mixed models to compare replicates using the library as a random effect (https://github.com/maragkakislab/nanoplen).

**Human data analysis**
**GTEx RNA-seq analysis.** Gene read counts for the brain (cortex), heart (left ventricle), and liver were obtained from the GTEx portal (V8 release). GTEx donors that experienced a slow death, defined by death after a long illness with >1 day of a terminal phase (DTHHRDY = 4), were filtered out. RNA-seq data was filtered for protein-coding genes with an average mRNA count ≥ 10. ImpulseDE2/0.99/10 [68] was employed in R/4.0.5 to identify transient and monotonous gene expression changes with age. The "vecConfounders" parameter was used to account for donor sex. An FDR-corrected $p$-value cutoff of 0.05 was used for statistical significance.

**Human 5hmC data.** Human 5hmC data for corresponding tissues were mined from published datasets, He et al.[31] and Cui et al.[30]. Tissue-

specific genes were obtained from the Human Protein Atlas and defined as genes with at least 4-fold higher expression in the tissue of interest compared to any other tissues ("tissue enriched") or the average of all other tissues ("tissue enhanced")[67]. BED files for the ImpulseDE2 differential genes and tissue-specific genes were generated using the University of California, Santa Cruz (UCSC) Genome Browser (Table Browser) with the appropriate genome assembly corresponding to the 5hmC data (hg38 for He et al.[31] and hg19 for Cui et al.[30]).

**Peng et al. RNA-seq and hMEDIP-seq data.** Matched RNA-seq and hMeDIP-seq data generated using T24 bladder cancer cells with and without 0.25 mM treatment with vitamin C was mined from Peng et al.[64] and processed similarly to mouse liver data except alignment was to the GRCh37/hg19 human reference genome.

**PCA plots.** hMeDIP-seq or MeDIP-seq PCA plots were generated using the plotPCA function of deepTools/3.5.0[95] using files obtained from the multiBigwigSummary function.

**Area under the curve (AUC) calculation.** To obtain genome coverage information (AUC) across regions of interest, we used the summary function of bwtool/1.0[106] with additional parameter "-with-sum".

**Annotation.** Genomic annotations were performed using annotatr/1.16.0[107] with "mm10_cpgs", "mm10_basicgenes", "mm10_genes_intergenic", and "mm10_genes_intronexonboundaries" as annotations.

**Gene ontology analysis.** GO analysis for the DHMRs and DMRs were performed using GREAT/4.0.4[41] with the mm10 genome as background and default association rule settings. The top 5 significant biological process category terms are reported (ranked according to FDR). For the oligo mass spec and GTEx data, GO analysis was performed using DAVID/6.8 with either *Mus musculus* (oligo mass spec) or *Homo sapiens* (GTEx) genes as background. The top biological process category terms with $p < 0.05$ are reported (ranked according to $p$-value).

**Genome browser tracks.** bigWig files for individual and pooled (across replicates) samples were used to generate genome browser tracks via the UCSC Genome Browser using either custom tracks or track hubs.

**Heatmaps.** All heatmaps were generated using the ComplexHeatmap/3.16 package in R with row z-score standardized values.

**Motifs.** Motif analysis to identify transcription factors was performed using Lisa (http://lisa.cistrome.org/).

**Hypergeometric test.** Hypergeometric tests for the venn diagrams were performed using EVenn[108].

**Reporting summary**
Further information on research design is available in the Nature Portfolio Reporting Summary linked to this article.

## Data availability
All genome-wide datasets generated in this study have been submitted to the Gene Expression Omnibus portal under GEO: GSE221124. Raw mass spec data are deposited at chorusproject.org/1795. hMeDIP and RNA-seq data of Vitamin C-treated T24 cells were obtained from Peng et al.[64] (https://doi.org/10.1186/s13148-018-0527-7). The MeDIP data is reported in Yang et al.[47] under GEO: GSE185708 [https://www.ncbi.nlm.nih.gov/geo/query/acc.cgi?acc=GSE223480]. Source data are provided with this publication and available at MendeleyData (https://doi.org/10.17632/mz5hgw2t4f.1). Source data are provided with this paper.

## Code availability

All code used in this study are available at GitHub (https://github.com/PSenlab/Occean_2024) and published in Zenodo (https://zenodo.org/doi/10.5281/zenodo.12167052)[109].

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

## Acknowledgements

We wish to acknowledge the National Institute on Aging Intramural Research Program (NIA IRP), National Institutes of Health (NIH), for financial support. This work was funded by grant NIH ZIA AG000679 to P.S. Y.S. and S.S. gratefully acknowledge the Leukemia Research Foundation, AFAR, the Japan Agency for Medical Research and Development, Merck, Deerfield, Einstein-Mount Sinai Pilot Diabetes grant, and NIH grants 1S10OD030286-01 and P30CA01333047. We thank the Comparative Medicine Section at NIA for providing support with animal experiments and Elin Lehrmann for GEO upload. We appreciate Myriam Gorospe, Weidong Wang and David Schlessinger, for critical reading of the manuscript. We thank Radhika Patnala and Arne Fabritius for illustrations. This work utilized the computational resources of the NIH HPC Biowulf cluster (http://hpc.nih.gov).

## Author contributions

P.S. conceptualized the project. J.O. and P.S. wrote the manuscript. J.O. performed wet lab experiments with help from N.Y., M. S. D., R.M., L.W. and C.Y.S. Y.S., J.S.K. and S.S. ran and analyzed DNA and oligo pulldown mass-spec data. N.Y. and C.B. performed direct RNA-seq experiments. S.D. and M.M. analyzed direct RNA-seq data. C.A. provided senescent cells and performed senescence measurements. J.O., N.Y. and P.S. performed bioinformatics analyses except direct RNA-seq. C.Y.C. assisted in DHE staining and mouse dissections. M.B. (Bernier), N.L.P. and R.dC. provided livers from DSF-treated animals. C.D. helped with TMRM flow cytometry assay. M.B. (Bhattacharyya) aided with oligo mass-spec experiments. J.F. and S.D. provided sequencing support.

## Funding

## Competing interests

The authors declare no competing interests.
