## [Peer Review File · Nature Communications]

Gene body DNA hydroxymethylation restricts the magnitude of transcriptional changes during agingREVIEWER COMMENTS

Reviewer #1 (Remarks to the Author):

In this manuscript, Ocean et al, using murine liver and cerebellum as model organs, reported 5hmC can restrict the variability and magnitude of gene expression during aging. This manuscript attempted to explain the role of 5hmC during aging and found the potential mechanism of 5hmC involved in aging. However, I have the following concerns and more data is needed to further support their biological findings and conclusion.

Major :

1. The authors clarified that "a significantly higher number of peaks were identified in the old (mean=126,934) compared to young (mean = 115,583)". However, the authors use different mapped fragments to call peaks, which may lead to different peak numbers. If you want to compare the number of peaks in different groups, please use the same sequencing reads.
2. The authors reported that lower dispersion and magnitude may be caused by gene expression levels (figure S1H). Have you ever compared the gene expression levels of the lower/higher fold change genes? The lower fold change genes, which may house keeping genes, may have higher gene expression levels and higher 5hmC levels. Thus, these gene expression levels are relatively stable. On the other hand, the higher fold change genes may have lower gene expression levels and you can find the 5hmC signals of this gene are lower.
3. Fig 3D/F showed higher 5hmC levels in the old, and the upstream or the downstream gene body also show higher 5hmC levels. This may be caused by globally increasing 5hmC levels or not normalizing very well. Please further illustrate the results. Moreover, these results only reflect the correlation between the 5hmC signals and age, the author should give more specific evidence to prove that 5hmC can restrict the variability and magnitude of gene expression changes with age.
4. Previous studies report that DNA methylation tends to increase with age (PMCID: PMC2847747; PMC6520108). This manuscript only showed a modest clustering of the samples by age (Supp. Fig. 3A), but the 5hmC signal can well distinguish the samples by age (Fig. 2A), which means that 5hmC is more indicative of age than 5mC?
5. In Figure S3B, the variation of the peaks of 5mC is too high. One sample only has about 2,500 peaks. It may be caused by a high duplication rate. If the duplication rate is much high, you should exclude failed samples for downstream analysis.
6. The authors reported the number of 5hmC peaks are 126,934(old) and 115,583(young). However, the number of 5hmC peaks are mean=32,435(old) and 32,435.5(young). As we all know, the content of 5mC is higher than 5hmC in the genome. Why did the authors find fewer 5mC peaks in the genome?
7. In Figure S4H, the two old samples are more similar to the two young samples than the old samples themselves. Page 13: "PCA plots of normalized transcript counts clearly segregated the samples by age". Can you explain that the samples were segregated by age?
8. The authors clarified that 5hmC accumulates at genic regions during aging in mouse, but they identified small parts of genes that showed upregulated with age in human tissues. Is there any difference between mouse and human?

Minor:

1, The order of the pictures is not marked in the order in which the articles appear. Such as Supp. Fig. 1H.

Reviewer #2 (Remarks to the Author):

The authors ascribe a role for hydroxymethylation predominately in the ageing liver. While the amount of work is impressive I remain unconvinced that a direct role between 5hmC and its ability to control gene expression exists. The basis of this finding seems to relate to figure 3B in which the gene body 5hmC is associated with decreased variance. Notably the authors point out that 5hmC is linked to genes that are highly expressed (5A) and they note:

“the low dispersion with high 5hmC could simply reflect the lower variance due to high expression and less noise in these genes. We later address this point by assigning a functional role for 5hmC in this process using liver regeneration as a method to reduce 5hmC and observing an outcome on transcriptional changes (see Fig 5)”

The authors point to figure 5 but I have been unable to determine how they have directly linked reduced 5hmC to an outcome on transcriptional change. I think the authors are referring here to the results in I, J and K? They report:

“the dilution of 5hmC after regeneration significantly increased the magnitude of transcriptional changes in genes that otherwise undergo low mRNA absolute fold change with age”

If the authors are trying to link the reduced 5hmC to increasing the change in logFC then they should separate the genes not by lower fold change in old/young but additionally based on the 5hmC levels. Based on Figure S1H there are a number of genes with low log fold change which have very low 5hmC signal. If these additionally show transcriptional response it is unlikely to be due to changes in 5hmC levels. Ideally the authors need to do further experiments in which they measure the 5hmC gene wide rather than globally and correlate a change in 5hmC with a change in transcription.

Some other comments:

1. How have the authors controlled for the global change in 5hmC levels when performing hMeDIP-seq?
2. MeDIP-seq has notorious bias (Down et al. Nat Biotechnology 2008) with relation to varying density of CpG dinucleotides. The authors have not reported correcting for this in any way. How have the authors addressed this?
3. The authors also should justify their choice of hMeDIP-seq over other experimental methods such as OxBS-seq, ACE-seq or nanopore sequencing.

Reviewer #3 (Remarks to the Author):

The Ocean et al manuscript reports several interesting findings centering around 5hmC. They showed

that 5hmC levels increase with age, particularly in the liver of mice, and also increased as cultured cells enter into senescence. They additionally suggested that 5hmC levels across gene body correlate with mRNA expression changes with aging. This latter point would need to be more thoroughly investigated (see below).

In general, the manuscript is dense, and the figure labels and figure legends are brief, making it difficult to read through.

e.g. "low mRNA fold change" is confusing. Is this referring to genes that show downregulated expression with aging? Overall, it'd be helpful to have clearer labels, more details in the figure legends.

Fig 2C – how was the significance determined? The data do not seem convincing. Same applied to Fig S6.

Fig 3D, left panel – do the data reflect the genes that show downregulation with age, i.e. the left group in Fig 3C? The writing was unclear.

3C-F – The analyses need to be redone to rule out that the pattern is due that lowly expressed genes are inherently more "noisy" due to detection limit. The authors can separate genes into low, medium, high expressed genes and conduct the analysis again to see if they continue to see a similar pattern. Similar comment for the cerebellum data.

Fig 3D, & F, right panels – Although statistically significant, the differences appear to be very small. The authors should try a different way of assessing the significance.

Fig 3F – is left vs right correspond to left vs right in 3E?

4F – It was concluded that higher 5hmC has higher level of alternative splicing. This also could be due limitation of alternative splicing might be easier to detect in genes that are higher expression and with high 5hmC.

Fig S3 & S7 - Relaxing threshold from FDR <0.05 to p <0.05 seems problematic. FDR should still be used even if the threshold is relaxed.

Reviewer #4 (Remarks to the Author):

This manuscript by Ocean et al. presents a large amount of data correlating 5hmC with a variety of aspects of gene structure, expression and processing. The causality of any of these correlations is not explored other than a reasonable demonstration that cellular quiescence, or lack thereof, is a determinant of the extent of 5hmC expression. Other than that many of the correlations and analysis are confusing at best. For example, related to Figure 4, line 263 mentioned 3 oligos but then only two are described (oligo 1 and 2), ribosomal proteins (e.g. RPS26 and 13) are labeled as splicing factors, data in Supp Fig 4D and E are both described as using proteins from old tissue but have different results (see line 280 and 286), and the comparisons in Fig 4D compare proteins from old tissues bound to 5hmC to proteins from young tissues bound to 5mC and C... so it is age of tissue or DNA modification that matters?

The authors would do much better to focus on one process they wish to connect to DNA hydroxymethylation (expression, gene structure, splicing, disease, etc) and carry out more clearly described analyses and experiments to test causality.

REVIEWER COMMENTS

We greatly appreciate the reviewers' comments on our manuscript (# NCOMMS-23-02305-T). We now present a revised version for your consideration.

Major edits:

1. We have switched from a peak-based analysis of MeDIP and hMeDIP datasets, to a genome-wide binning analysis using QSEA (quantitative sequencing enrichment analysis). QSEA corrects for CpG density per bin, offering a more comprehensive estimate of enrichment differences between young and old.
2. To assess the causal relationship between 5hmC and transcription restriction, we have performed additional experiments and analyses, summarized in new Fig. 6:

(a) hMeDIP-seq and RNA-seq on regenerated liver from young and old mice to test the effect of regeneration-induced **reduction of 5hmC** on transcription restriction. Note, we have used livers from the *same* animal to evaluate the effect of regeneration (pre- and post-surgery) to eliminate any biological variations.

(b) hMeDIP-seq and RNA-seq of T24 bladder cancer cells +/- vitamin C (Peng et al, *Clinical Epigenetics*, 2018) to test the effect of **increased 5hmC levels** on transcription restriction.

(c) **Additional analyses** to clarify 5hmC's role in response to reviewer comments (detailed point-by-point below).

Minor edits:

1. Added duplication rates for meDIP/hmeDIP libraries in new Supp. Figs. 1G, 2G and 6G.
2. Revised Figs. 2, S3 and S7 to remove peak-based analyses and replace with QSEA analyses.
3. Change schematic in Fig. 4A to better describe oligo mass-spec experiments.
4. Replaced volcano plots in Fig. 4B-C and Supp. Fig. 4D-G.
5. We removed some figure panels (for example original Figs. 3I, S8D) to accommodate new ones and for sake of clarity.
6. Changed color theme from black and red to black and salmon.
7. Textual changes (tracked) to improve readability.
8. Expanded figure legends to improve clarity.
9. Changes to Supplementary Tables to incorporate changes made during revision.

Our overall conclusions remain consistent with our first submission:

1. We find that 5hmC is stably enriched in multiple aged organs. In the aged mouse liver, the increase of 5hmC is driven by prolonged quiescence and senescence.
2. The age-related accumulation of 5hmC occurs over gene bodies and, importantly, over genes associated with tissue-specific functions. In the case of the liver, this occurs primarily over nuclear-encoded metabolic and mitochondrial genes.
3. Gene body 5hmC functions as a transcriptional rheostat during aging by restricting the magnitude of transcriptional changes with age. In other words, having 5hmC on a gene reduces its likelihood of high expression change with age, which we call "transcription restriction".
4. 5hmC regulates transcription restriction through decreased binding to splicing-associated factors, compared to unmodified cytosine and 5mC.
5. We verify that 5hmC has a causal role in transcription restriction by showing that reduction of 5hmC by liver regeneration modulates the magnitude of transcriptional changes with age.
6. Lastly, we found evidence for 5hmC's transcription restriction function in mouse cerebellum and human tissues, and further show that this restriction reduces tissue-specific function and is detrimental in the face of stress. 5hmC is altered by regimens known to regulate lifespan (high-fat diet) and recent drug treatments (disulfiram). We propose that modulating 5hmC levels may play a role in promoting longevity.

New findings from revision:

Our new experiments aimed to manipulate 5hmC levels and test transcriptional outcomes.

1. We *reduced 5hmC levels* over gene bodies by liver regeneration and found that it increases the transcriptional magnitude of genes that otherwise showed minimal change with age.
2. We *increased 5hmC levels* over gene bodies by vitamin C treatment (by reanalyzing published data from Peng et al) and found that it reduces the transcriptional magnitude of genes that gain 5hmC.

We provide a point-by-point response to reviewer comments **in red** below.

Reviewer #1 (Remarks to the Author):

In this manuscript, Occean et al, using murine liver and cerebellum as model organs, reported 5hmC can restrict the variability and magnitude of gene expression during aging. This manuscript attempted to explain the role of 5hmC during aging and found the potential mechanism of 5hmC involved in aging. However, I have the following concerns and more data is needed to further support their biological findings and conclusion.

Major:

1, The authors clarified that “a significantly higher number of peaks were identified in the old (mean=126,934) compared to young (mean = 115,583)”. However, the authors use different mapped fragments to call peaks, which may lead to different peak numbers. If you want to compare the number of peaks in different groups, please use the same sequencing reads.

Response

We thank the reviewer for this comment. In our original submission, we performed MACS2 peak calling on the individual immunoprecipitated (IP) samples, i.e., young or old (using the corresponding input sample as control) to identify peaks. We state that the mean **number** of peaks (vs background) that was identified for old was greater than young (original Fig. 2B). Comparing total number of peaks might convey important information about modification abundance across biological samples provided the experiments were done under similar conditions, normalized to sequencing depth, and statistically assessed across biological replicates which we did (see Methods). Importantly, these are not differentially enriched peaks. The peaks can be at different genomic locations in young and old as the reviewer rightly points out.

To compare peak enrichments from the same genomic location, we initially performed a differential analysis using the tool DiffBind (original Fig. 2E) which compares peak signals across young and old from the same genomic location. This differential analysis gives more quantitative comparisons between samples (original Fig. 2E-G).

Nevertheless, as pointed out by Reviewer #2, meDIP-seq has been shown to be CpG density-biased^{1,2}. Therefore, in the revised version of the manuscript, we have switched from the typical peak analysis to another differential enrichment tool (QSEA or quantitative sequencing enrichment analysis³) that normalizes for CpG density (new Figs. 2, S3 and S7). We have therefore removed mentions of MACS2, peak number, and DiffBind altogether from the revised manuscript as QSEA bins the genome and does not require peak calling.

2, The authors reported that lower dispersion and magnitude may be caused by gene expression levels (figure S1H). Have you ever compared the gene expression levels of the lower/higher fold change genes?

The lower fold change genes, which may house keeping genes, may have higher gene expression levels and higher 5hmC levels. Thus, these gene expression levels are relatively stable. On the other hand, the higher fold change genes may have lower gene expression levels and you can find the 5hmC signals of this gene are lower.

Response

Please see Fig. R1 below for the mRNA levels of genes with low and high absolute expression fold change (renamed minimal and maximal change respectively in response to Reviewer 3’s comments) with age. We (and the reviewer) rightly speculate that genes in the maximal change group (right panel) had lower median mRNA levels in both young and old samples compared to genes in the minimal change group (left panel). Thus, lowly expressed genes, which are marked by low levels of 5hmC, may be more dispersed or “noisy”, and, therefore, show a more exaggerated difference in absolute fold change between young and old. Conversely, highly expressed genes are marked by high 5hmC and therefore show less noise and lower absolute fold changes. However, as noted by the individual data points, each group shows a wide range of expression, containing genes with “very low” and “very high” mRNA counts.

Fig. R1: mRNA levels of genes with minimal and maximal expression change with age

(A) Boxplot overlaid with individual data points showing log₂mRNA counts of genes with minimal change (left panel) or maximal change (right panel) in expression with age. Note both plots are on the same scale and the median for genes with minimal change with age is higher than those maximal change. *** = p < 0.001 by Welch’s t-test.

It is difficult to uncouple 5hmC levels and transcription since the two are positively correlated, i.e., genes with higher expression (left panel) have higher 5hmC and vice versa (also noted in Figs. 3A and Supp. Fig. 8A). The same relationship with transcription is also observed for H3K36me3⁴ and H3K4me3 breadth⁵. We address the causality of 5hmC's impact on transcription restriction with additional analyses and experiments (Figs. R2, R7, new Fig. 6).

Of relevance to this reviewer's point, we divided all genes into five groups based on either gene body 5hmC levels (Fig. R2A) or absolute mRNA fold change (Fig. R2B-C). We observe in both cases, the negative relationship between 5hmC levels and magnitude of transcriptional changes with age is not just evident at extreme values but rather progressive across the different groups.

Fig. R2: Relationship between 5hmC and mRNA fold change

(A) Genes divided into 5 groups based on increasing 5hmC in the old. The *absolute* mRNA fold change is plotted in the y-axis and shows a progressive decrease with increasing 5hmC signal in the old. (B) Genes divided into 5 groups based on increasing *absolute* mRNA fold change. (C) The corresponding 5hmC levels of the 5 groups in (B) are traced as metaplots. As with (A), (B-C) show a progressive decrease in 5hmC levels with increasing *absolute* mRNA fold change with age.

3. Fig 3D/F showed higher 5hmC levels in the old, and the upstream or the downstream gene body also show higher 5hmC levels. This may be caused by globally increasing 5hmC levels or not normalizing very well. Please further illustrate the results.

Response

We understand the reviewer's concern but local effects such as the one pointed out by the reviewer are not uncommon for modifications that show high signal. We believe that this effect is not due to any global differences. IP-sequencing experiments (such as hMeDIP-seq, MeDIP-seq or CHIP-seq) do not capture global differences unlike assays such as western or dot blots. This is because after IP, we make libraries from equal amount (ng) of pulldown DNA. We further carefully measure nM concentrations of the libraries using Qubit and qPCR methods before loading equal amount (nmoles) on the Illumina sequencer. This "equalizing" abrogates global differences and only local differences are apparent. Equal loading of our samples is evident in Fig. R3A, which shows there were no significant differences in the FASTQ file sizes between young and old.

We also do not believe this is a normalization issue as we followed ENCODE guidelines including duplicate removal, removal of blacklist regions, seq-depth, and input normalization in our study (see associated github for detailed code). We present the following 2 evidence to strengthen our argument that the normalization was adequate:

- (a) Genes with higher fold change (renamed upregulated (Fig. 3D, right panel)) or higher *absolute* fold change (renamed maximal change (Fig. 3F, right panel)) do not show a marked difference upstream and downstream of the gene body. This group represents approximately the same number of genes as those with lower fold change (renamed

downregulated (Fig. 3D, left panel)) or lower *absolute* fold change (renamed minimal change (Fig. 3F, left panel)).
 (b) We further binned the genome into 1Mb bins and plotted *raw* 5hmC signal (Fig. R3B), *seq-depth normalized* 5hmC signal (Fig. R3C) and *seq-depth + input normalized* 5hmC signal (Fig. R3D) across the genome for both young and old. There were significant differences between young and old samples in the raw signal (Fig. R3B) and with seq-depth normalization (Fig. R3C), but after input normalization, the differences were erased. Since we used *seq depth + input normalized signal* for all our assessments, we conclude that our samples were normalized adequately and that any evident local differences (such as in Figs. 3D & 3F) are true.

Fig. R3: Evidence of successful normalization of hMeDIP samples

(A) Boxplot of FASTQ file sizes generated from young and old hMeDIP datasets. (B) Boxplot with dots showing individual area under the curve (AUC) raw measurements for 1Mb bins genome-wide for each young and old sample. (C) Same as B except plotted with AUC values after seq-depth normalization. (D) Same as B except plotted with AUC values after seq-depth + input normalization. For (A-D), significance was assessed using ANOVA. *p*-values are reported on the top. *N.S.* is not significant.

Moreover, these results only reflect the correlation between the 5hmC signals and age, the author should give more specific evidence to prove that 5hmC can restrict the variability and magnitude of gene expression changes with age.

Response

We agree with the reviewer. We now present additional experiments and analyses (new Fig. 6) to prove that 5hmC *causally* affects transcriptional restriction. We address causality in two ways:

- (1) by *decreasing* 5hmC by liver regeneration, and
- (2) by *increasing* 5hmC by vitamin C treatment and then measuring mRNA absolute fold changes.

#1 was achieved by 70% partial hepatectomy surgeries on young and old mice. Note, we used livers from the same animal to evaluate the effect of regeneration (pre- and post-surgery) to eliminate any biological variations. #2 was achieved by mining data from Peng et al, in which case T24 bladder cancer cells were treated with vitamin C. In both cases, we analyzed matched hMeDIP-seq and RNA-seq datasets. We refrained from TET enzyme manipulations to change 5hmC levels because none of the 3 TETs had changed expression with age (Fig. 5A-B).

Our results revealed two important conclusions:

- (a) We found that manipulating 5hmC levels did not change transcription levels (Fig. R4). In other words, the positive correlation between gene body 5hmC and transcription (Figs. 3A and Supp. Fig. 8A) is primarily due to transcription driving gene body 5hmC levels higher rather than vice versa, as previously proposed^{6,7}.

(b) We found that decreasing 5hmC levels by regeneration increased *absolute* fold change of the corresponding mRNAs (new Fig. 6E & 6F). Conversely, increasing 5hmC levels by vitamin C decreased *absolute* fold change of the mRNAs in bladder cancer cell lines (new Fig. 6J). These results are now summarized in new Fig. 6 and described in the text in this revision. Overall, these new findings establish a more direct influence of 5hmC on transcription restriction.

Fig. R4: 5hmC manipulations don't affect transcription

(A) Boxplot overlaid with individual data points showing log₂mRNA counts of genes with minimal change (left panel) or maximal change (right panel) in expression in old pre-surgery and post-surgery samples (i.e., before and after regeneration). (B) Boxplot overlaid with individual data points showing log₂mRNA counts of genes with minimal change (left panel) or maximal change (right panel) in expression in control or vitamin C-treated T24 cells (data from Peng et al). For (A-B), *** = $p < 0.001$ and N/S = $p \geq 0.05$ by Welch's t-test. (C) Metaplots of 5hmC signal over bodies of genes with low ($n = 6307$), intermediate ($n = 6306$), and high ($n = 6306$) fold change between old post-surgery and old pre-surgery liver samples (i.e., before and after regeneration). (D) Metaplots of 5hmC signal over the bodies of genes with low ($n = 5605$), intermediate ($n = 5604$), and high ($n = 5604$) fold change between control and vitamin C-treated T24 cells (data from Peng et al). Note, for (C-D), there is no relationship between 5hmC signal and transcription in the post-surgery/vitamin C-treated samples.

4. Previous studies report that DNA methylation tends to increase with age (PMCID: PMC2847747; PMC6520108). This manuscript only showed a modest clustering of the samples by age (Supp. Fig. 3A), but the 5hmC signal can well distinguish the samples by age (Fig. 2A), which means that 5hmC is more indicative of age than 5mC?

Response

We thank the reviewer for this question. 5mC differences with age/senescence/age-associated cancer are partially driven by hypermethylation at PRC2 targets (CpG promoters)⁸⁻¹⁰ and partially by hypomethylation at lamin-associated domains (LADs)¹¹⁻¹⁴. Since LADs occupy a sizeable portion of the genome, the hypomethylation pattern is more likely to be a top distinguishing variable in PCA plots using genome-wide coverage such as in Fig. 2A or Supp. Fig. 3A.

In our data (reported in Yang et al, *Mol. Cell*, 2023¹⁰), we did not observe global hypomethylation at LADs (Fig. 2G and 2Q in Yang et al), likely because we used 18-month-old mice. However, epigenetic drift is predicted to be stronger at later ages.

MeDIP is also inherently biased toward genomic regions with low CpG density¹ which could technically preclude information from CpG-rich promoters where age-related hypermethylation occurs. To overcome this drawback of MeDIP, we mined published methyl-CpG binding domain (MBD) protein followed by sequencing (MBD-seq) data from Mozhui et

al¹⁵ from young (4 months), middle (11-12 months) and old (24 months) mouse livers. MBD-seq is also an affinity-based enrichment technique but unlike MeDIP, is biased towards capturing regions with high CpG density¹. The mined MBD-seq data also did not show any strong separation by age in C57BL/6J mice. (Fig. R5). In conclusion, 5mC genome coverage patterns up to 24 months of age, do not seem to be very different between young and old.

Fig. R5: Analysis of MBD-seq data from young and old mouse livers
(A) PCA plot showing genome coverage of 5mC using mined MBD-seq data from Mozhui et al¹⁵.

In contrast to MeDIP, our hMeDIP-seq showed that the 5hmC genome coverage could distinguish young and old samples (Fig. 2A). Comparing 5hmC and 5mC might be challenging due to potential confounders; antibodies (anti-5mC vs anti-5hmC), and enrichment at different genomic loci. Therefore, we refrain from stating that 5hmC maybe a superior indicator of age. Instead, we simply say that in genic regions 5hmC changes are more apparent between young and old (measured using hMeDIP-seq, new Fig. 2G) but not 5mC (measured by MeDIP-seq, new Supp. Fig. 3G).

5. *In Figure S3B, the variation of the peaks of 5mC is too high. One sample only has about 2,500 peaks. It may be caused by a high duplication rate. If the duplication rate is much high, you should exclude failed samples for downstream analysis.*

Response

We thank the reviewer for pointing this out. We also noted that 5mC had high variability in the number of peaks identified by MACS2. As QC, we have assessed the duplication rates and observed similar duplication rates among all the samples (duplication rates have now been added for all Me/hMeDIP experiments as new Supp. Figs. 1G, 2G and 6G). Also shown in Supp. Fig. 2C-F, there were no significant differences in sequencing depth, mapped fragments, or fragment length. We additionally performed a FRiP (Fraction Reads in Peaks) score analysis which estimates the % of reads that fall within peaks and is used by ENCODE to assess data quality. We noted that FRiP scores for consensus peaks across all samples in the MeDIP dataset were $\geq 4\%$ (Table R1). By ENCODE standards, FRiP scores above 1% are considered successful. Thus, we believe the low peak number in the young 1 sample (2500 peaks) is likely a biological phenomenon.

Regardless, as mentioned in our response to this reviewer's question 1 and reviewer 2 comments, we have removed peak-based analysis in this revision and performed QSEA which is an enrichment-based analysis that also considers differences in CpG differences. In conclusion, we believe that are our data passes all quality checks and that any differences observed are likely biological.

6. *The authors reported the number of 5hmC peaks are 126,934(old) and 115,583(young). However, the number of 5hmC peaks are mean=32,435(old) and 32,435.5(young). As we all know, the content of 5mC is higher than 5hmC in the genome. Why did the authors find fewer 5mC peaks in the genome?*

Response

Both meDIP and hmeDIP-seq are affinity enrichment-based assays that greatly depend on antibody specificity, binding affinity, and IP efficiency. Overall, peak numbers cannot be directly compared across MeDIP and hMeDIP (only relative comparisons within the same experiment across age groups can be assessed).

In response to this reviewer, we confirmed that our pull-down efficiencies in each assay reflected a successful experiment by measuring Fraction of Reads in Peaks (FRiP) scores which by ENCODE guidelines should be more than 1%. Our average FriP scores for liver hMeDIP were 25.5%, for cerebellum hMeDIP was 19.5% and for liver MeDIP was 4.13% (Table R1). We also confirmed the specificity of the pull-downs in Supp. Figs. 1B, 2B and 6B. Altogether, our MeDIP and hMeDIP assays were successful but notably hMeDIP experiments had higher FRiP scores possibly indicating that the anti-5hmC antibody was better and therefore yielded more peaks.

Alternatively, 5mC may have more broader enrichments (rather than peaks) across the genome compared with 5hmC, that cannot be captured by peak calling algorithms like MACS2. In this situation, there would also be a discrepancy in the number of peaks called on both modifications.

In contrast to affinity-based methods, mass spectrometric detection of DNA modifications presents an antibody-free quantitative way to better estimate relative abundances. Our mass spec data indeed indicates that 5mC is globally higher than 5hmC (Fig. 1A).

Finally, as we mentioned before, in the revised version of the manuscript we have switched from the typical peak analysis to another differential coverage tool, QSEA³ and therefore we have now removed peak comparisons throughout the manuscript.

7, *In Figure S4H, the two old samples are more similar to the two young samples than the old samples themselves. Page 13: "PCA plots of normalized transcript counts clearly segregated the samples by age". Can you explain that the samples were segregated by age?*

Response

The reviewer is correct. We had both sexes in this study, and we noted that PC1 is sex while PC2 is age. The young and old samples that the reviewer points out were close in PCA were both from males. We have now edited the text to specify that PC1 segregated the samples by sex while PC2 segregated the samples by age.

8, *The authors clarified that 5hmC accumulates at genic regions during aging in mouse, but they identified small parts of genes that showed upregulated with age in human tissues. Is there any difference between mouse and human?*

Response

We thank the reviewer for this comment and agree this needs to be clarified. While GTEx provides a large dataset of RNA-seq in aging human subjects, 5hmC assessments in human samples are few. Additionally, there is high variability in human samples due to genetics, lifestyle, health status etc. Thus, we felt we were underpowered to, and did not, perform an old vs young analysis of 5hmC genic signal from the limited data. We are currently planning experiments to expand the existing 5hmC datasets in human samples which would more definitively answer this question.

Minor:

1, *The order of the pictures is not marked in the order in which the articles appear. Such as Supp. Fig. 1H.*

Response

We double checked the order of the figures in the article and believe that all our figures are cited in order. For Supp. Fig. 1H (now Supp. Fig. 1I), we have updated the legend to describe all 4 panels.

Reviewer #2 (Remarks to the Author):

The authors ascribe a role for hydroxymethylation predominately in the ageing liver. While the amount of work is impressive I remain unconvinced that a direct role between 5hmC and its ability to control gene expression exists. The basis of this finding seems to relate to figure 3B in which the gene body 5hmC is associated with decreased variance. Notably the authors point out that 5hmC is linked to genes that are highly expressed (5A) and they note:

"the low dispersion with high 5hmC could simply reflect the lower variance due to high expression and less noise in these genes. We later address this point by assigning a functional role for 5hmC in this process using liver regeneration as a method to reduce 5hmC and observing an outcome on transcriptional changes (see Fig 5)"

The authors point to figure 5 but I have been unable to determine how they have directly linked reduced 5hmC to an outcome on transcriptional change. I think the authors are referring here to the results in I, J and K? They report:

"the dilution of 5hmC after regeneration significantly increased the magnitude of transcriptional changes in genes that otherwise undergo low mRNA absolute fold change with age"

If the authors are trying to link the reduced 5hmC to increasing the change in logFC then they should separate the genes not by lower fold change in old/young but additionally based on the 5hmC levels.

Response

We greatly appreciate this reviewer's comments (also raised by reviewer 1 points 2 and 3) and for identifying the key figures that attempt to assign a role for 5hmC and transcription restriction in our original submission. To address all the abovementioned points and determine a causal relationship between 5hmC and transcriptional restriction we have

performed two experiments (and additional analyses) in this revision:

- (a) we decreased 5hmC by liver regeneration,
- (b) we increased 5hmC by vitamin C treatment (achieved by mining data from Peng et al, where T24 bladder cancer cells were treated with vitamin C)

and then measuring the magnitude in gene expression changes. In both cases, we analyzed matched hMeDIP-seq and RNA-seq datasets. Decreasing 5hmC levels by regeneration increased *absolute* fold change of the corresponding mRNAs (new Fig. 6E & 6F). Conversely, increasing 5hmC levels by vitamin C decreased *absolute* fold change of the mRNAs in bladder cancer cell lines (new Fig. 6J). These results are now summarized in new Fig. 6 and described in the text in this revision.

As to the point the reviewer makes about ranking by 5hmC, after performing the hMeDIP-seq in the regenerated samples, we observed that 5hmC is essentially obliterated in both age groups after regeneration in all genes (new Fig. 6C). We therefore cannot meaningfully separate genes based on 5hmC levels after regeneration to estimate mRNA fold changes. Rather, we rank genes based on 5hmC signal in the pre-surgery condition (new Fig. 6F). Additionally, we restrict our analysis to look at the gene group that had high 5hmC pre-surgery and minimal expression change with age (new Fig. 6E). In both cases, we observe that reducing 5hmC levels *increases* transcriptional amplitude of these genes, confirming 5hmC's role in directly regulating the magnitude in gene expression changes with age.

Based on Figure S1H there are a number of genes with low log fold change which have very low 5hmC signal. If these additionally show transcriptional response it is unlikely to be due to changes in 5hmC levels. Ideally the authors need to do further experiments in which they measure the 5hmC gene wide rather than globally and correlate a change in 5hmC with a change in transcription.

Response

We appreciate this perspective of the reviewer. To test this, we took all genes with mean 5hmC signal below 0 in Fig. S1H (new Fig. S1I) panels 1 and 2 for both young and old and then plotted the absolute mRNA fold changes before and after regeneration. As shown in Fig. R6 below, they were not significantly different between the two groups. In other words, since this gene set already had very low 5hmC levels to begin with, liver regeneration which further dilutes 5hmC, did not have any effect on transcriptional magnitudes.

Fig. R6: Absolute mRNA fold changes of genes harboring very low 5hmC signal

(A) Boxplot overlaid with individual data points showing absolute mRNA fold changes for genes with very low (mean signal < 0) 5hmC on gene bodies. Fold changes are calculated as old pre-surgery vs young pre-surgery group (for before regeneration) and old post-surgery (240h) vs young pre-surgery group (for after regeneration). N/S = $p \geq 0.05$ by Mann-Whitney U test.

Some other comments:

1. How have the authors controlled for the global change in 5hmC levels when performing hMeDIP-seq?

Response

We thank the reviewer for this question. We elaborated on this in response to Reviewer 1's point #3. Briefly, IP-sequencing

experiments (such as hMeDIP-seq, MeDIP-seq or ChIP-seq) do not capture global differences unlike assays such as western or dot blots. This is because after IP, we make libraries from equal amount (ng) of pulldown DNA. We further carefully measure nM concentrations of the libraries using Qubit and qPCR methods before loading equal amount (nmoles) on the sequencing machine. This “equalizing” abrogates global differences and only local differences are apparent. Equal loading of our samples is evident in Fig. R3A which shows there were no significant differences in the FASTQ file sizes between young and old. Therefore, while we show 5hmC is globally high by dot blot (Fig. 1B) and immunofluorescence (Fig. 1C), in hMeDIP-seq analysis, we are only measuring local differences.

2. *MeDIP-seq has notorious bias (Down et al. Nat Biotechnology 2008) with relation to varying density of CpG dinucleotides. The authors have not reported correcting for this in any way. How have the authors addressed this?*

Response

We agree with the reviewer regarding the CpG-density bias associated with MeDIP-seq, which tends to favor regions with lower CpG density¹. Consequently, peaks identified using MACS2 in our previous approach might exhibit a bias toward regions with low CpG density. While this could have limited our comprehensive view of 5hmC distribution in the genome *within* sample, we believe that age-related differential peaks obtained should remain unaffected by varying CpG densities, as this value is region-specific and expected to be consistent *across* old and young samples. Additionally, one limitation of adopting a differential analysis approach covering the entire genome, rather than focusing solely on relatively enriched regions captured by MeDIP-seq, might potentially reduce the detection of genuine age-related changes. This shift arises from the increased stringency of multiple testing correction applied in analyzing a broader genomic landscape, leading to a more stringent statistical threshold.

Nevertheless, to address this limitation of MeDIP-seq, we have now transitioned to using QSEA (quantitative sequencing enrichment analysis)³ as the method for differential enrichment analysis. Unlike tools that rely on peaks, QSEA partitions the genome into user-defined bins to evaluate statistical differences in enrichment between two groups. This new approach enables age-related comparisons across all bins/regions, irrespective of CpG density. Additionally, using a sigmoidal CpG density bias curve, QSEA accounts for CpG density-dependent enrichment per bin in the differential analysis, offering a more comprehensive and unbiased assessment of 5hmC distribution across the entire genome. QSEA is now employed for all differential analyses in our revised manuscript involving hMeDIP-seq and MeDIP-seq data (new Fig. 2, and new Supp. Figs. 3 and 7). The re-analyzed data affirms our conclusion that age-related accumulation of 5hmC primarily occurs in gene bodies of tissue-specific genes (new Fig. 2F, and new Supp. Figs. 3F and 7F).

3. *The authors also should justify their choice of hMeDIP-seq over other experimental methods such as OxBS-seq, ACE-seq or nanopore sequencing.*

Response

We greatly appreciate the reviewer's question. Our choice of hMeDIP-seq was motivated by the fact that it is an established and cost-effective assay that directly measures 5hmC.

We did not use OxBS-seq for the following reasons:

- (a) OxBS-seq estimates 5hmC levels by subtraction which we thought would be an indirect measure for our purposes.
- (b) A key limitation of oxBS-seq (or any bisulfite-based method) is the use of bisulfite, as chemical deamination conditions can degrade as much as 99.9% of input genomic DNA (gDNA). We thought it was important to ensure the integrity of our sample as gDNA from old livers may already be damaged and hence be prone to degradation.
- (c) Sequencing cost – we ask the reviewer to consider that we are a new lab, the cost per sample for oxBS-seq was beyond our budget.

ACE-seq is a relatively new method that overcomes the challenges of chemical deamination by using enzymatic deamination with APOBEC3A (A3A). A3A readily deaminates C and 5mC, but exhibits a ~5000-fold reduction in the 5hmC deamination rate relative to that of C. The main challenges with ACE-seq are:

- (a) A3A is not commercially available, and all steps of the process require to be carefully optimized in our system. Importantly, A3A amounts would have to be titrated to ensure full C/5mC deamination in all sequence contexts.
- (b) APOBEC enzymes exhibit sequence context preferences, leading to variable conversion rates in different sequence contexts.
- (c) Like oxBS-seq, ACE-seq requires a much higher sequencing depth for reliable 5hmC calling, driving up costs.
- (d) Finally, ACE-seq data requires extensive computational analysis to distinguish true 5hmC signals from potential false positives. To the best of our knowledge a public computational pipeline for this method is not available.

At the time we performed these experiments (2020), there were several impediments to nanopore-sequencing and 5hmC detection:

- (a) There was no established software to reliably identify and distinguish 5hmC modifications from 5mC.
- (b) Even the few experimental tools that were under development had no published performance metrics (and still don't).
- (c) All tools were trained exclusively with human data and could not be generalized to other species.

These limitations with nanopore are best summarized in Liu et al, *Genome Biology*, 2021¹⁶. Citing a relevant section of the paper:

“However, very few computational methods are available that can predict 5hmC from nanopore reads. SignalAlign—a three-way (C, 5mC, or 5hmC) cytosine classifier trained by synthetic oligonucleotides—achieved an accuracy of 79% for predicting cytosine with 5hmC, but the method is developed with nanopore chemical version R7.3 (the pore is out-of-date and no longer available) and its repository has not been updated for over 4 years. We also noticed that ONT recently published a “research release” on basecalling model trained in 5hmC and 5mC in all contexts in the Rerio repository, and Megalodon will be able to predict 5hmCs and 5mCs simultaneously. However, the 5hmC model is still under development, and to date no data are available on its performance to predict 5hmC.”

Considering the challenges of the alternatives, hMeDIP, to us, was the most straightforward assay to detect 5hmC.

Reviewer #3 (Remarks to the Author):

The Occean et al manuscript reports several interesting findings centering around 5hmC. They showed that 5hmC levels increase with age, particularly in the liver of mice, and also increased as cultured cells enter into senescence. They additionally suggested that 5hmC levels across gene body correlate with mRNA expression changes with aging. This latter point would need to be more thoroughly investigated (see below).

In general, the manuscript is dense, and the figure labels and figure legends are brief, making it difficult to read through. e.g. “low mRNA fold change” is confusing. Is this referring to genes that show downregulated expression with aging? Overall, it'd be helpful to have clearer labels, more details in the figure legends.

Response

We are sorry that the reviewer found it difficult to read the manuscript. For clarity, we have now rephrased some of the confusing terms, expanded figure legends, and added clearer labels to figures. We have also removed some figure panels to reduce crowdedness in the figures. The reviewer is right in that low mRNA fold change indeed refers to genes that are downregulated with age. For clarity we have replaced some terms:

- (a) “low mRNA fold change” is now “downregulated”
- (b) “high mRNA fold change” is now “upregulated”
- (c) “low *absolute* mRNA fold change” is now “minimal change”
- (d) “high *absolute* fold change is now “maximal change”

We have also revised the text, legends, and corresponding figure panels to reflect this nomenclature.

Fig 2C – how was the significance determined? The data do not seem convincing. Same applied to Fig S6.

Response

For the supplementary figures, we assume the reviewer is referring to Supp. Figs. 3C and 7C. We used Welch's t-test and classified significance by the conventional $p < 0.05$. Our significant p -value was likely due to the large number of data points. Large sample sizes tend to have more statistical power to detect minor but significant differences. However, in this revised manuscript, we have switched from a peak-based analyses to QSEA (based on Reviewer 2's comments), and therefore these panels have been removed in this revised manuscript.

Fig 3D, left panel – do the data reflect the genes that show downregulation with age, i.e. the left group in Fig 3C? The writing was unclear.

Response

The reviewer is correct. The left panel in Fig. 3D indeed shows the 5hmC profiles of the genes “downregulated” in expression with age. We have clarified this in the text, legend, and figure panel.

3C-F – The analyses need to be redone to rule out that the pattern is due that lowly expressed genes are inherently more “noisy” due to detection limit. The authors can separate genes into low, medium, high expressed genes and conduct the

analysis again to see if they continue to see a similar pattern. Similar comment for the cerebellum data.

Response

We thank the reviewer for this comment. In the original manuscript, we grouped the genes based on low, medium, and high fold change and reported only the low and high groups (Fig. 3C-F). In response similar comments from Reviewer 1, we performed a more extensive binning and separated the genes by *both* 5hmC and *absolute* mRNA fold change into 5 groups. We note that the relationship between 5hmC and gene expression change during aging is progressive and not just limited to the lowly expressed genes. Please see response to Reviewer 1 (point 2) and Fig. R2 for details.

Fig 3D, & F, right panels – Although statistically significant, the differences appear to be very small. The authors should try a different way of assessing the significance.

Response

We completely agree with the reviewer that the differences are small in Fig. 3D and 3F (right panels). In the original manuscript we performed Welch's t-test (parametric test) to measure significance. We have now also assessed using Mann-Whitney U test, a non-parametric test, which still shows a significant difference for both 3D and 3F (right panels). Since we do not focus on this minor difference in the manuscript, we retained the original test but now clarify in the text that the effect size is very small.

Fig 3F – is left vs right correspond to left vs right in 3E?

Response

Yes, that is correct. We have now added arrows for clarity.

4F – It was concluded that higher 5hmC has higher level of alternative splicing. This also could be due limitation of alternative splicing might be easier to detect in genes that are higher expression and with high 5hmC.

Response

We thank the reviewer for this comment and acknowledge that since splicing is intimately linked to transcription, which in turn is linked to 5hmC, our conclusion may simply be a readout of these relationships. It is well known in literature that transcription and 5hmC are positively correlated (also shown in Figs. 3A and Supp. Fig. 8A). It has also been shown that TET proteins travel with elongating RNAPII to oxidize 5mC to 5hmC^{6,7}. Therefore, we believe transcription increases 5hmC. However, the opposite may not necessarily be true, i.e., 5hmC may not necessarily increase transcription. We test this in Fig. R4 where we found no relationship between 5hmC levels (either decreased by liver regeneration or increased by vitamin C treatment, data from Peng et al¹⁷) and transcription levels. With this premise, we address this reviewer's point by measuring alternative splicing under conditions of low and high 5hmC and without any direct effects on transcription. We observe that despite no transcriptional effects, various alternative splicing events were affected. In regenerated livers, which had a reduction of 5hmC over gene bodies that show minimal expression change with age (new Fig. 6D, left panel), alternative splicing events were partially restored to youthful levels (Fig. R7A). This is in agreement with the disfavored binding of splicing factors by 5hmC in our oligo mass-spec studies (Fig. 4) which leads to more alternative splicing events. In contrast, vitamin C treatment of T24 cells led to an increase in 5hmC over gene bodies that show minimal expression change with age (new Fig. 6I, bottom left panel) and in turn showed increased alternative splicing events (Fig. R6B).

Nevertheless, we do not rule out that in our aged liver data (Fig. 4F), detection of higher levels of alternative splicing over genes that undergo minimal expression change with age, may have partially stemmed from the ease of detection in highly transcribed genes.

Fig. R7: Manipulating 5hmC levels by liver regeneration and vitamin C treatment alters splicing

(A) Heatmap of normalized inclusion levels of all alternative splicing events by rMATS in young, old pre-surgery (before regeneration) and old post-surgery (after regeneration) liver samples. **(B)** Number of differential splicing events grouped by minimal or maximal expression change in the control and vitamin C-treated T24 bladder cancer cells from Peng et al.

Fig S3 & S7 - Relaxing threshold from FDR <0.05 to $p < 0.05$ seems problematic. FDR should still be used even if the threshold is relaxed.

Response

As mentioned before, we switched from a peak-based analysis to QSEA in this revised manuscript. For MeDIP analysis presented in Supp. Fig. 3, no differential regions survived FDR of 0.05. For hMeDIP analysis of cerebellum in Supp. Fig. 7, very few regions (n=74) survived FDR of 0.05. To enable *some* functional interpretation (such as GO analysis), we used $p < 0.05$. However, we acknowledge in the text that this may point to potential false positives. For sake of transparency, we provide all the FDR values for the QSEA differential regions in Supp. Table S3 and S10. Additionally, we emphasize that neither 5mC in liver nor 5hmC in the cerebellum show strong age-related changes.

Reviewer #4 (Remarks to the Author):

This manuscript by Occean et al. presents a large amount of data correlating 5hmC with a variety of aspects of gene structure, expression and processing. The causality of any of these correlations is not explored other than a reasonable demonstration that cellular quiescence, or lack thereof, is a determinant of the extent of 5hmC expression. Other than that many of the correlations and analysis are confusing at best. For example, related to Figure 4, line 263 mentioned 3 oligos but then only two are described oligo 1 and 2),

Response

We thank the reviewer for their feedback and acknowledge the concerns. The causality concern was raised by reviewers 1-3 as well and therefore we have performed additional experiments/analyses. Briefly:

- (a) we *decreased* 5hmC by liver regeneration,
- (b) we *increased* 5hmC by vitamin C treatment (achieved by mining data from Peng et al, where T24 bladder cancer cells were treated with vitamin C)

and then measured transcriptional outcomes. In both cases, we analyzed matched hMeDIP-seq and RNA-seq datasets. Decreasing 5hmC levels by regeneration increased the magnitude of expression changes of the corresponding genes (new Fig. 6E & 6F). Conversely, increasing 5hmC levels by vitamin C decreased the magnitude of gene expression changes of the corresponding genes in bladder cancer cell lines (new Fig. 6J). These results are now summarized in new Fig. 6 and described in the text in the revised manuscript.

With regards to the question about the oligos, we agree that we need to clarify the text better. We used 2 different oligos (oligo 1 and oligo 2) designed from 2 different genomic regions. Each of these 2 oligos were further modified in 3 ways as follows:

- (a) Unmethylated (where all Cs are unmodified)
- (b) Methylated (where all Cs are 5mC modified)
- (c) Hydroxymethylated (where all Cs are 5hmC modified)

Therefore, our experiments were performed with $2 \times 3 = 6$ different oligos. We have now clarified this in the text updated

the schematic (new Fig. 4A) to avoid confusion.

ribosomal proteins (e.g. RPS26 and 13) are labeled as splicing factors,

Response

We thank the reviewer for pointing out this discrepancy. We used the set of proteins that were significantly enriched or de-enriched in our oligo mass-spec dataset and ran a pathway analysis using DAVID GO. From this GO analysis, we saw that most of the terms were related to chromatin, RNA splicing, translation, and metabolism. In Fig. 4B-C, we simply tried to color the significantly interacting proteins using specific GO terms related to these 4 categories. The specific RNA splicing related GO term we used was GO:0008380 (RNA splicing) and while both *RPS26* and *RPS13* are included in this GO term (under negative regulation of RNA splicing, subterm GO:0033119), we agree that these proteins are not classical splicing factors but rather indirectly related to splicing. To avoid any confusion, we have now revised the volcano plot (new Fig. 4B-C and new Supp. Fig. 4D-G). We removed categorization and instead highlighted specific proteins (known splicing factors on the de-enriched side).

data in Supp Fig 4D and E are both described as using proteins from old tissue but have different results (see line 280 and 286), and the comparisons in Fig 4D compare proteins from old tissues bound to 5hmC to proteins from young tissues bound to 5mC and C... so it is age of tissue or DNA modification that matters?

Response

It was important for us to compare the enrichment profiles in old 5hmC to *old* 5mC/C as well as *young* 5mC/C. This is because we believe that *both* the modification and aging may simultaneously contribute to differences in bound proteins. For example:

- (a) old 5hmC vs *old* mC/C comparison will identify 5hmC specific proteins in the old lysate (new Supp. Fig. 4D & 4F).
- (b) old 5hmC vs *young* mC/C comparison will identify 5hmC specific proteins, different from (a), maybe due to age-related changes in protein abundance (new Supp. Fig. 4E & 4G).

The authors would do much better to focus on one process they wish to connect to DNA hydroxymethylation (expression, gene structure, splicing, disease, etc) and carry out more clearly described analyses and experiments to test causality.

Response

We are disappointed that the reviewer found our manuscript unfocused. Our key findings relate 5hmC to transcription restriction during aging. We think that presenting the correlations between 5hmC, transcription, quiescence (aging is essentially prolonged quiescence¹⁰) and interventions that alter 5hmC levels are important to understand it's complexity during aging. We hope our revisions (including causality experiments) clarify this reviewer's concerns.

References

1. Nair, S.S. *et al.* Comparison of methyl-DNA immunoprecipitation (MeDIP) and methyl-CpG binding domain (MBD) protein capture for genome-wide DNA methylation analysis reveal CpG sequence coverage bias. *Epigenetics* **6**, 34-44 (2011), 10.4161/epi.6.1.13313.
2. Down, T.A. *et al.* A Bayesian deconvolution strategy for immunoprecipitation-based DNA methylome analysis. *Nat Biotechnol* **26**, 779-85 (2008), 10.1038/nbt1414.
3. Lienhard, M. *et al.* QSEA-modelling of genome-wide DNA methylation from sequencing enrichment experiments. *Nucleic Acids Res* **45**, e44 (2017), 10.1093/nar/gkw1193.
4. Pu, M. *et al.* Trimethylation of Lys36 on H3 restricts gene expression change during aging and impacts life span. *Genes Dev* **29**, 718-31 (2015), 10.1101/gad.254144.114.
5. Benayoun, B.A. *et al.* H3K4me3 breadth is linked to cell identity and transcriptional consistency. *Cell* **158**, 673-88 (2014), 10.1016/j.cell.2014.06.027.
6. Laird, A., Thomson, J.P., Harrison, D.J. & Meehan, R.R. 5-hydroxymethylcytosine profiling as an indicator of cellular state. *Epigenomics* **5**, 655-669 (2013).
7. Pfeifer, G.P. & Szabó, P.E. Gene body profiles of 5-hydroxymethylcytosine: potential origin, function and use as a cancer biomarker. *Epigenomics* **10**, 1029-1032 (2018), 10.2217/epi-2018-0066.
8. Horvath, S. *et al.* DNA methylation clocks tick in naked mole rats but queens age more slowly than nonbreeders. *Nat Aging* **2**, 46-59 (2022), 10.1038/s43587-021-00152-1.
9. Moqri, M. *et al.* PRC2 clock: a universal epigenetic biomarker of aging and rejuvenation. *bioRxiv*, 2022.06.03.494609 (2022), 10.1101/2022.06.03.494609.

10. Yang, N. *et al.* A hyper-quiescent chromatin state formed during aging is reversed by regeneration. *Mol Cell* **83**, 1659-1676 e11 (2023), 10.1016/j.molcel.2023.04.005.
11. Cruickshanks, H.A. *et al.* Senescent cells harbour features of the cancer epigenome. *Nat Cell Biol* **15**, 1495-506 (2013), 10.1038/ncb2879.
12. Lochs, S.J.A., Kefalopoulou, S. & Kind, J. Lamina Associated Domains and Gene Regulation in Development and Cancer. *Cells* **8**(2019), 10.3390/cells8030271.
13. Hon, G.C. *et al.* Global DNA hypomethylation coupled to repressive chromatin domain formation and gene silencing in breast cancer. *Genome Res* **22**, 246-58 (2012), 10.1101/gr.125872.111.
14. Brinkman, A.B. *et al.* Partially methylated domains are hypervariable in breast cancer and fuel widespread CpG island hypermethylation. *Nat Commun* **10**, 1749 (2019), 10.1038/s41467-019-09828-0.
15. Mozhui, K. & Pandey, A.K. Conserved effect of aging on DNA methylation and association with EZH2 polycomb protein in mice and humans. *Mech Ageing Dev* **162**, 27-37 (2017), 10.1016/j.mad.2017.02.006.
16. Liu, Y. *et al.* DNA methylation-calling tools for Oxford Nanopore sequencing: a survey and human epigenome-wide evaluation. *Genome Biol* **22**, 295 (2021), 10.1186/s13059-021-02510-z.
17. Peng, D. *et al.* Vitamin C increases 5-hydroxymethylcytosine level and inhibits the growth of bladder cancer. *Clin Epigenetics* **10**, 94 (2018), 10.1186/s13148-018-0527-7.

REVIEWER COMMENTS

Reviewer #1 (Remarks to the Author):

In this response, Ocean et al. resolved almost all questions. However, before I can recommend the publication of the work in Nature Communications, there are still some key points that should be clarified.

1. The response to the second question remains insufficient in addressing the query. It is evident that genes exhibiting minimal change with age tend to have higher gene expression. Consequently, a more substantial alteration in gene expression is required to manifest a greater fold change. Conversely, genes displaying maximal change with age exhibit lower overall gene expression, thus supporting the expectation of a larger change. Therefore, the utilization of fold change as a basis for comparison in this context may not be deemed reasonable.
2. On page 4, "We now present additional experiments and analyses (new Fig. 6) to prove that 5hmC causally affects transcriptional restriction." Does this statement contradict "We found that manipulating 5hmC levels did not change transcription levels"?
3. The authors failed to indicate the revised sections, rendering it challenging to locate the alterations within the text.

Reviewer #3 (Remarks to the Author):

In response to the previous reviewers comments, the authors employed a new strategy to analyze their data. The authors provided a thorough set of analyses and responses to the reviewers. I believe the findings presented in the paper provided support to the conclusions the authors put forward, and the experiments and data represent a thorough and rigorous effort based on available technologies. Overall, I believe the findings presented in this paper are valuable to the field.

Reviewer #4 (Remarks to the Author):

the authors have added much new data to the manuscript which significantly strengthen the study and support the conclusions.

REVIEWER COMMENTS

We greatly appreciate the reviewers' comments on our manuscript (# NCOMMS-23-02305-T). We now present our response to Reviewer 1 for consideration.

Edits in the manuscript to this version:

We have included a limitation section (at the end of Discussion) to indicate that we could not perform sex-specific analyses of age-related differences due to limited sample size.

Our overall conclusions remain consistent with our first submission and subsequent revision:

1. We find that 5hmC is stably enriched in multiple aged organs. In the aged mouse liver, the increase of 5hmC is driven by prolonged quiescence and senescence.
2. The age-related accumulation of 5hmC occurs over gene bodies and, importantly, over genes associated with tissue-specific functions. In the case of the liver, this occurs primarily over nuclear-encoded metabolic and mitochondrial genes.
3. Gene body 5hmC functions as a transcriptional rheostat during aging by restricting the magnitude of transcriptional changes with age. In other words, having 5hmC on a gene reduces its likelihood of high expression change with age, which we call "transcription restriction".
4. 5hmC regulates transcription restriction through decreased binding to splicing-associated factors, compared to unmodified cytosine and 5mC.
5. We verify that 5hmC has a causal role in transcription restriction by showing that reduction of 5hmC by liver regeneration modulates the magnitude of transcriptional changes with age.
6. Lastly, we found evidence for 5hmC's transcription restriction function in mouse cerebellum and human tissues, and further show that this restriction reduces tissue-specific function and is detrimental in the face of stress. 5hmC is altered by regimens known to regulate lifespan (high-fat diet) and recent drug treatments (disulfiram). We propose that modulating 5hmC levels may play a role in promoting longevity.
7. In new experiments during the first revision we aimed to manipulate 5hmC levels and test transcriptional outcomes.
 - a. We *reduced 5hmC levels* over gene bodies by liver regeneration and found that it increases the transcriptional magnitude of genes that otherwise showed minimal change with age.
 - b. We *increased 5hmC levels* over gene bodies by vitamin C treatment (by reanalyzing published data from Peng et al) and found that it reduces the transcriptional magnitude of genes that gain 5hmC.

We provide a point-by-point response to reviewer comments **in red** below.

Reviewer #1 (Remarks to the Author):

In this response, Occean et al. resolved almost all questions. However, before I can recommend the publication of the work in Nature Communications, there are still some key points that should be clarified.

1. The response to the second question remains insufficient in addressing the query. It is evident that genes exhibiting minimal change with age tend to have higher gene expression. Consequently, a more substantial alteration in gene expression is required to manifest a greater fold change. Conversely, genes displaying maximal change with age exhibit lower overall gene expression, thus supporting the expectation of a larger change. Therefore, the utilization of fold change as a basis for comparison in this context may not be deemed reasonable.

We acknowledge the reviewer's concerns about gene expression levels potentially confounding the relationship between 5hmC and the observed changes in age-related mRNA expression, a point we have recognized in both the original manuscript and in the first revision of the manuscript (revision 1, page 8, lines 195-200). Despite the challenge in decoupling these factors, we have undertaken multiple experiments and analyses to address this issue.

In our previous rebuttal, we demonstrated that the association between 5hmC levels and the magnitude of age-related changes in gene expression persists when genes are categorized by gene body 5hmC levels rather than by mRNA fold change (revision 1, Fig. R2A). Moreover, using a liver regeneration model to decrease 5hmC levels (Fig. 6A-E), we found that genes exhibiting **minimal changes with age** under "normal conditions" showed significantly greater changes in mRNA expression in the reduced 5hmC conditions (Fig. 6E). Notably, these comparisons **involve the same genes**, and **the same reference group** (young pre-surgery) with consistent expression levels, thereby eliminating gene expression levels as a confounding factor.

In the latest revision of our manuscript, we have added the label “genes with minimal change” above the plot in Fig. 6E to highlight the specific group of genes under comparison. Overall, we believe these results provide compelling evidence for a more direct influence of 5hmC on restricting age-related transcriptional changes.

2. On page 4, "We now present additional experiments and analyses (new Fig. 6) to prove that 5hmC causally affects transcriptional restriction." Does this statement contradict "We found that manipulating 5hmC levels did not change transcription levels"?

We appreciate the reviewer's observation of a perceived contradiction and welcome the opportunity to provide further clarification. An important detail we omitted in the quoted sentence, but discussed in the subsequent sentence, concerns the direction of the effect between gene body 5hmC and transcription levels. In the response, we clarified that “in other words, the positive correlation between gene body 5hmC and transcription (Figs. 3A and Supp. Fig. 8A) is primarily due to transcription driving up gene body 5hmC levels rather than the reverse, as previously proposed^{6,7}”. Our intention was to convey that, although 5hmC is positively associated with gene expression levels, it is not necessarily a positive regulator of transcription but rather a consequence. This model was initially proposed by Laird et al. (2013) and Pfeifer (2018) and is supported by our findings that 5hmC serves to moderate rather than enhance/upregulate transcription. We apologize for any confusion caused and acknowledge that our statement “manipulating 5hmC levels did not change transcription levels” could have been more succinctly phrased to reflect this understanding.

3. The authors failed to indicate the revised sections, rendering it challenging to locate the alterations within the text.

We apologize profusely for this. We uploaded a tracked word document but perhaps the pdf conversion erased the tracking. In this revision, we have converted our tracked word to pdf before uploading.

Reviewer #3 (Remarks to the Author):

In response to the previous reviewers comments, the authors employed a new strategy to analyze their data. The authors provided a thorough set of analyses and responses to the reviewers. I believe the findings presented in the paper provided support to the conclusions the authors put forward, and the experiments and data represent a thorough and rigorous effort based on available technologies. Overall, I believe the findings presented in this paper are valuable to the field.

We thank the reviewer for their kind comments and support of the manuscript.

Reviewer #4 (Remarks to the Author):

The authors have added much new data to the manuscript which significantly strengthen the study and support the conclusions.

We are grateful to this reviewer as well for their critical feedback and suggestions that have significantly improved our manuscript.

REVIEWERS' COMMENTS

Reviewer #1 (Remarks to the Author):

The authors have further clarified the issues and eliminated ambiguous statements, resulting in a text that more effectively supports the study's conclusions.